# European daily precipitation according to EURO-CORDEX RCMs and high-resolution GCMs from HighResMIP

Marie-Estelle Demory[1], Ségolène Berthou[2], Jesus Fernandez[3], Silje L. Sørland[1], Roman Brogli[1], Malcolm J. Roberts[2], Urs Beyerle[1], Jon Seddon[2], Rein Haarsma[4], Christoph Schär[1], Erasmo Buonomo[2], Ole B. Christensen[5], James M. Ciarlò[6], Rowan Fealy[7], Grigory Nikulin[8], Daniele Peano[9], Dian Putrasahan[10], Christopher D. Roberts[11], Retish Senan[11], Christian Steger[12], Claas Teichmann[13], Robert Vautard[14]

[1] Institute for Atmospheric and Climate Science, ETH, 8092 Zürich, Switzerland
[2] Met Office Hadley Centre, EX1 3PB Exeter, United Kingdom
[3] University of Cantabria, 39005 Santander, Spain
[4] KNMI, 3731 GA De Bilt, Netherlands
[5] Danish Meteorological Institute (DMI), 2100 Copenhagen, Denmark
[6] The Abdus Salam International Center for Theoretical Physics (ICTP), 34135 Trieste, Italy; and the National Institute of Oceanography and Applied Geophysics (OGS), 34010 Sgonico, Italy
[7] ICARUS/Department of Geography, Maynooth University, Maynooth, Co. Kildare, Ireland
[8] Swedish Meteorological and Hydrological Institute (SMHI), 60176 Norrköping, Sweden
[9] CMCC, 40127 Bologna, Italy
[10] Max-Planck-Institut für Meteorologie (MPI), 20146 Hamburg, Germany
[11] ECMWF, RG2 9AX Reading, United Kingdom
[12] Deutscher Wetterdienst (DWD), 63067 Offenbach, Germany
[13] Climate Service Center Germany (GERICS), Helmholtz-Zentrum Geesthacht, 20095 Hamburg, Germany
[14] Institut Pierre-Simon Laplace, Paris, France

*Correspondence to*: Marie-Estelle Demory (marie-estelle.demory@env.ethz.ch)

**Abstract.** In this study, we evaluate a set of high-resolution (25-50 km horizontal grid spacing) Global Climate Models (GCMs) from the High-Resolution Model Intercomparison Project (HighResMIP), developed as part of the EU-funded PRIMAVERA project, and the EURO-CORDEX Regional Climate Models (RCMs) (12-50 km horizontal grid spacing) over a European domain. It is the first time that an assessment of regional climate information using ensembles of both GCMs and RCMs at similar horizontal resolutions has been possible. The focus of the evaluation is on the distribution of daily precipitation at 50km scale under current climate conditions. Both the GCM and RCM ensembles are evaluated against high-quality gridded observations in terms of spatial resolution and station density. We show that both ensembles outperform GCMs from the 5th Coupled Model Intercomparison Project (CMIP5), which cannot capture the regional-scale precipitation distribution properly because of their coarse resolutions. PRIMAVERA GCMs generally simulate precipitation distributions within the range of EURO-CORDEX RCMs. Both ensembles perform better in summer and autumn in most European regions, but tend to overestimate precipitation in winter and spring. PRIMAVERA shows improvements in the latter by reducing moderate-precipitation rate biases over Central and Western Europe. The spatial distribution of mean precipitation is also

improved in PRIMAVERA. Finally, heavy precipitation simulated by PRIMAVERA agrees better with observations in most regions and seasons while CORDEX overestimates precipitation extremes. However, uncertainty exists in the observations due to a potential undercatch error, especially during heavy precipitation events.

The analyses also confirm previous findings that, although the spatial representation of precipitation is improved, the effect of increasing resolution from 50 km to 12 km horizontal grid spacing in EURO-CORDEX daily precipitation distributions is, in comparison, small in most regions and seasons outside mountainous regions and coastal regions. Our results show that both high-resolution GCMs and CORDEX RCMs provide adequate information to end-users at 50km scale.

## 1 Introduction

Climate models are essential tools to provide information on the evolution of climate quantities, their variability, and interactions with various components of the Earth System. They are designed to balance model resolution, physics complexity and computational requirements. Therefore, depending on the requirements of the experiment (e.g. size of computational domain, length of the simulated period, number of realizations), only a relatively coarse spatial resolution can be afforded. There have been two main streams of development in the climate modelling community: Global Climate Models (GCMs) and Regional Climate Models (RCMs). The latter were developed to alleviate the computational burden of GCMs by focusing on a particular region, where higher spatial resolution (about one order of magnitude higher) can be achieved with the same computational power (Laprise, 2008; Giorgi, 2019). GCMs are complex models that account for interactions at the global scale between various components of the Earth System (e.g. atmosphere, ocean, sea ice, vegetation). RCMs are complex models that dynamically downscale GCM results to obtain finer climate information for a particular region. Both approaches have complementary advantages and drawbacks, as summarised by Denis et al, 2002: 1) For a given computational power, RCMs reach higher horizontal resolution than state-of-the-art GCMs over the region of interest. As a result, RCMs provide a more detailed representation of complex topography and land-sea contrast (e.g. Torma et al., 2015). While GCMs are able to provide climate information at the global scale, RCMs provide the tools for process understanding, particularly related to small-scale processes and extreme events (e.g. Diaconescu and Laprise, 2013; Ban et al., 2015). 2) Sub-grid physical processes are represented by parameterization schemes that have been developed for the resolutions of the RCM (12-50 km horizontal grid spacing). At such resolution, these schemes may be more appropriate than GCM schemes that are developed at much coarser resolutions (100-300 km horizontal grid spacing) (e.g. Giorgi and Mearns, 1999; Prein et al., 2016). 3) RCMs parameterization

schemes can be chosen based on the appropriateness for the region, and tuning can be completed to better match regional observations (e.g. Bellprat et al., 2016), while it is not possible to apply region-specific tuning in GCMs (Mauritsen et al., 2012; Hourdin et al., 2017). 4) RCMs introduce a new source of uncertainty, since they require lateral boundary conditions from GCMs and, therefore, cannot fully replace them (Denis et al., 2002; Diaconescu and Laprise, 2013). Spatial domain size and location introduce additional sources of uncertainty. Small domains cannot spin up higher-resolution features (e.g., Brisson et al., 2016), while domains that are too large can create unrealistic circulation patterns (Prein et al., 2019). Conversely, high-resolution GCMs alone have the potential to provide regionally-relevant climate information globally. 5) RCMs can downscale various GCMs to sample many different large-scale climate conditions at the domain boundaries. This ability to provide large ensembles is an important step to evaluate the RCM ensemble spread and better constrain modelling uncertainties. However, the decision on which GCMs are downscaled can be subjective and affect the uncertainty range (Fernandez et al., 2019). 6) Global Earth System Models (such as those used for the Coupled Model Intercomparison Projects, CMIP5 and CMIP6) are usually more complex and include more climate components than state-of-the-art RCMs. The simpler treatment of several climate components in RCMs (e.g. aerosols, vegetation) has been shown to artificially reduce the uncertainty range of their driving GCMs (Boé et al., 2020). Regional Earth System Models are developed to counteract this drawback (Turuncoglu and Sannino, 2017; Giorgi, 2019; Zhang et al., 2020). While high-resolution GCMs will present significant new opportunities, the ability to employ regionally specific parameterisation schemes at ever high spatial resolutions means that RCMs will remain essential tools to supplement global Earth System Models.

**1.1 CORDEX RCMs**

Since the end of the 1980s, dynamical downscaling has been used to provide regional climate projections (Dickinson et al., 1989; Giorgi, 2019), and has become a well-accepted and extensively used approach to produce climate change information at the regional scale (refer to various national climate assessment reports, e.g. Kjellström et al., 2016; Fealy et al., 2018; Fernandez et al., 2019; Sørland et al., submitted; Nationaler Klimareport (https://www.dwd.de/DE/leistungen/nationalerklimareport/report.html); ReKliEs-De (http://reklies.hlnug.de); UK Climate Projections (https://www.metoffice.gov.uk/research/approach/collaboration/ukcp/index)). The Coordinated Regional Climate Downscaling Experiment (CORDEX) is an international coordinated effort to produce multi-model regional climate change projections (Giorgi et al., 2009; Gutowski et al., 2016). The main goal of CORDEX, which commenced in 2009, is to develop a framework that provides consistent high-resolution climate information at the regional scale to complement the information provided by GCMs. By systematically evaluating regional climate downscaling techniques, it aims to provide a solid scientific basis for impact assessments. Last but not least, it aims to promote interaction and communication between the Global Climate Modelling community, the Regional Climate Modelling community, and end users to better support adaptation activities (Giorgi et al., 2009).

The CORDEX initiative (Giorgi et al., 2009) has primarily focused its efforts on downscaling CMIP5 GCMs (150-200 km horizontal grid spacing) using dynamical downscaling techniques at a common 50 km (CORDEX-44) grid spacing. As

computational resources have become more available, horizontal grid spacings in RCMs have been further refined to 12 km (CORDEX-11) over Europe (e.g. Jacob et al., 2014; Kotlarski et al., 2014; Jacob et al., 2020), and a full GCM-RCM simulation matrix has been completed through the EU Copernicus Climate Change Services (C3S) PRINCIPLES (Producing Regional Climate Projections Leading to European Services) (Vautard et al., in revision; Coppola et al., in revision). Over other domains, the horizontal grid spacing has been refined to 25 km (CORDEX-22). These horizontal grid spacings were chosen as a compromise between what is computationally possible for the majority of modeling groups and the expected added value compared to GCMs. These community efforts within CORDEX have proved fruitful in providing reliable climate information in terms of mean and extreme temperature, precipitation, or wind (e.g. Kotlarski et al., 2014; Prein et al., 2016; Glisan et al., 2019), as well as their projected climate change signals over different parts of the globe (e.g. Gao et al., 2008; Jacob et al., 2014; Rajczak and Schär, 2017). Overall, CORDEX RCMs have been shown to improve the representation of mean climate compared to their driving GCMs, particularly evident over complex terrain associated with their higher resolution (Torma et al., 2015; Giorgi et al., 2016; Sørland et al., 2018). For example, when the RCM horizontal grid spacing is refined from 50 km to 12 km, a number of authors have found an improvement in terms of spatial and temporal distributions, particularly in mean and extreme precipitation in mountainous regions (Torma et al., 2015; Prein et al., 2016) due to its improved representation of orography. Summer seasons also tend to be better simulated in CORDEX-11 because the larger scales of convection are captured by the better resolved-scale dynamics (Prein et al., 2016). In addition, CORDEX-11 also shows improvement over CORDEX-44 in simulating amplitudes and historical trends of extreme autumn precipitation events over the Mediterranean coast, which have increased in intensity about 20% in the past 60 years (Luu et al., 2018). However, there is no clear benefit in going from 50 km (CORDEX-44) to 12 km (CORDEX-11) regarding mean climate and variability (e.g. Kotlarski et al., 2014; Casanueva et al., 2016b; Jury et al., 2019).

At grid spacings of both 50 or 12 km, the quality of RCM simulations has been shown to be linked to the internal skill of the RCM itself, which can be assessed by evaluating reanalysis-driven simulations (e.g. Kotlarski et al., 2014), but also to the quality of their driving GCMs (Giorgi and Mearns, 1999; Rummukainen, 2010; Diaconescu and Laprise, 2013; Hall, 2014). In mid-latitudes, particularly in the winter season, RCMs are strongly constrained by the large-scale mean circulation of the GCMs, which means that the RCM response is sensitive to the GCM circulation biases (Hall, 2014; Kjellström et al., 2016; Brogli et al., 2019; Fernandez et al., 2019), although RCMs tend to correct some of the biases evident in their driving GCMs (e.g. Guo and Wang, 2016; Sørland et al., 2018). In other regions, for example in the Tropics, or in other seasons, local-scale processes can be more important than large-scale drivers and the quality of RCM simulations is therefore less dependent on the driving GCMs. For example, RCMs have demonstrated ability to simulate summer convective precipitation extremes, which largely contribute to regional water budgets (e.g. Prein et al., 2016). In contrast, biases in simulated radiation and surface wind speeds appear to be more related to the RCM than the driving GCM (Vautard et al., in revision).

## 1.2 High-resolution GCMs

GCMs have developed in terms of complexity, particularly through the incorporation of new Earth System components. Over the past decade, GCM horizontal resolution has also increased, typically from ~300 km for CMIP3 models which provided the basis for the Intergovernmental Panel on Climate Change's (IPCC) Fourth Assessment Report (AR4) (Randall et al., 2007), to ~150 km for CMIP5, employed in the Fifth Assessment report (AR5) (Flato et al., 2013). CMIP6 (Eyring et al., 2016), which will provide the basis for the Sixth Assessment Report (AR6), have recently been completed at ~100 km horizontal grid spacing. Reflecting these model developments, a new high-resolution model intercomparison project, HighResMIP (Haarsma et al., 2016), has recently emerged. HighResMIP provides an evaluation framework for atmosphere-only and coupled GCM simulations at horizontal grid spacings of 50-25 km, in addition to simulations at more standard horizontal resolutions, to understand the role of increasing horizontal resolution in simulations of global climate mean, variability and extremes. HighResMIP simulations have recently finished and analyses are currently underway. Until recently, the effects of increasing horizontal resolution in GCMs have been investigated in a non-coordinated way and typically with a single or small number of models (e.g. Jung et al., 2012; Kinter III et al., 2013; Bacmeister et al., 2014; Mizielinski et al., 2014; Wehner et al., 2014). They have drawn similar conclusions regarding the interactions (or feedbacks) between the synoptic scales and the large-scale global climate system. For example, increasing horizontal resolution in GCMs plays a role in the simulation of the global hydrological cycle (Roberts et al., 2018b), which tends to be more intense but partitioned more realistically over land and ocean compared to observations due to stronger transport of atmospheric moisture (Demory et al., 2014) and a better representation of orography (Vannière et al., 2019). Coupling the atmosphere with ocean eddy-permitting models tends to improve the climate mean state and variability (e.g. Minobe et al., 2008; Shaffrey et al., 2009; Roberts et al., 2016). Synoptic-scale dynamics are better resolved in GCMs with increasing resolution, which improves the representation of mid-latitude eddy-driven jet variability, extra-tropical cyclones and associated extreme precipitation (Catto et al., 2010; Haarsma et al., 2013; Schiemann et al., 2018; Baker et al., 2019), as well as blocking events (Matsueda and Palmer, 2011; Berckmans et al., 2013). Intensity of tropical cyclones in GCMs also increases with resolution, and their interannual variability is better captured (e.g. Zhao et al., 2009; Roberts et al., 2015), but the resolution in GCMs is still not high enough to capture the most intense tropical cyclones. All these synoptic-scale processes can affect regional climate variability (e.g. Haarsma et al., 2013); their improved simulation can potentially lead to more realistic climate information and climate change projections at the regional scale (e.g. Matsueda and Palmer, 2011). This question is particularly important in regions where the water budget is partly driven by synoptic systems, such as tropical cyclones over East Asia (e.g. Guo et al., 2017) and Central America (e.g. Franco-Diaz et al., 2019) and frontal systems and eddy-driven jet interactions with topography over Europe (e.g. Woollings et al., 2010; Catto et al., 2012; Baker et al., 2019).

In this study, we make use of the available RCM and GCM coordinated efforts (CORDEX, CMIP, HighResMIP) to investigate the level of information given by various products, whether they are from low-resolution GCMs (CMIP5), high-resolution GCMs (HighResMIP), low-resolution EURO-CORDEX RCMs (EUR-44) and high-resolution EURO-CORDEX RCMs

(EUR-11). We would like to determine if the HighResMIP GCMs provide information at the regional scale that is comparable to regional climate CORDEX simulations. In other words, is the potential improvement of large-scale drivers of European climate with high-resolution GCMs as beneficial as the local tuning of regional models? This would enable us to inform end users about the kind of information they can expect by considering different products. We focus our efforts on the daily precipitation distribution over European regions at 50 km scale under current climate conditions. Section 2 presents the data used as well as the method employed to evaluate the daily precipitation distribution. Section 3 presents results of seasonal mean spatial simulation of precipitation and daily precipitation distribution. Section 4 includes sensitivity tests of the method and discusses the robustness of our results regarding EUR-11 versus EUR-44, and the uncertainty in observations. The main conclusions of this study are drawn and discussed in Section 5.

## 2 Method and data

### 2.1 PRIMAVERA GCMs

We use the ocean-atmosphere coupled GCMs developed and run within the EU-Horizon 2020 PRIMAVERA project (https://www.primavera-h2020.eu), which is a European contribution to HighResMIP. PRIMAVERA uses the HighResMIP protocol (Haarsma et al., 2016), which is different from CMIP (e.g. different aerosols; refer to Haarsma et al., 2016, for details). We use the PRIMAVERA simulations which were available at the time of the study (6 GCMs, see Table 1). Most high-resolution simulations include one member only, but in case there are more (IFS-HR provides 6 members), we consider one member per model in order to apply equal weight to each model (Table 1).

### 2.2 CORDEX RCMs

Over Europe, we use the CMIP5-driven EUR-44 and EUR-11 CORDEX simulations run at 0.44° (about 50 km) and 0.11° (about 12 km) horizontal grid spacings (Tables 2 and S1). Daily precipitation model data have been extracted from the Earth System Grid Federation (ESGF) servers. The advantage of considering both EUR-44 and EUR-11 is that EUR-44 RCMs horizontal grid spacing roughly corresponds to that of PRIMAVERA GCMs, and EUR-11 is based on state-of-the-art model generations (EUR-44 is slightly older). These two EURO-CORDEX ensembles make the comparison with state-of-the-art high-resolution GCM simulations more appropriate. As for PRIMAVERA, we use one member per GCM-RCM pair (Table S1).

It is worth noting that despite the similar grid spacing in EUR-44 and PRIMAVERA models, the effective resolution (Skamarock 2004; Klaver et al., 2020) of these models might differ considerably depending on their numerical integration scheme and energy dissipation mechanisms.

## 2.3 CMIP5 GCMs

To investigate the added value of CORDEX and PRIMAVERA simulations to CMIP5 GCMs, we constrain our study to the subset of CMIP5 GCMs used to force CORDEX simulations (Table 2, second column), available on the ESGF servers. However, we examine the robustness of our findings by also analysing the entire ensemble of CMIP5 simulations. Considering the full set changes the ensemble spread (not shown) but the main conclusions of the study regarding CMIP5 remain the same.

We perform our analysis either on the full CORDEX and PRIMAVERA ensembles or on reduced ensembles. Reduced ensembles correspond to PRIMAVERA GCMs and CORDEX RCMs that downscale CMIP5 GCMs based on the same GCM family, for example PRIMAVERA MPI-ESM1-2-XR and EUR-44 RCA4, CCLM4, CCLM5 and REMO2009 that downscale MPI-ESM-LR (blue colored in Table 2).

## 2.4 Observations

Over Europe, we make use of high spatial resolution gridded observational datasets that include the highest station density (Fig. S1): SAFRAN-V2 (France; Vidal et al., 2010), UKCPobs (British Isles; Perry et al., 2009), ALPS-EURO4M (Alps; Isotta et al., 2014), CARPATCLIM (Carpathian region; Szalai et al.). To cover the Iberian Peninsula, we combine Spain02 v2 (Herrera et al., 2012) and PT02 v2 (Belo-Pereira et al., 2011). For other regions, we considered E-OBS v17 (Cornes et al., 2018). E-OBS uses the complete observational stations network for Scandinavia, Netherlands, Germany (Gerard van der Schrier, personal communication). E-OBS is therefore expected to be of good quality over these regions. For the remaining regions, such as the Mediterranean region and Eastern Europe, we also make use of E-OBS, although the quality is most likely lower there (Prein and Gobiet, 2017). All the observation datasets used are listed in Table 3. Pros and cons of using such observational datasets are discussed in Sect. 4.3.

## 2.5 Period

To match the observation time periods with PRIMAVERA and CORDEX ensembles, we focus our analyses on the 1971-2005 historical period over Europe.

## 2.6 Domains

We divide the European domain into subregions according to the areas covered by the observational datasets (Fig. S1). Over the sub-regions covered by E-OBS, we consider the PRUDENCE regions (Christensen and Christensen, 2007). Throughout the paper, we therefore use AL for the Alps, BI for British Isles, FR for France, CA for Carpathians, CE for Central Europe, IP for Iberian Peninsula, MD for Mediterranean basin, NEE for Northeast Europe, SC for Scandinavia.

## 2.7 Description of precipitation distribution analysis

The daily precipitation distribution in each sub-region (Fig. S1) was assessed using a method similar to Berthou et al. (2019), based on the ASoP1 diagnostics tool developed by Klingaman et al. (2017). We calculate the daily precipitation distribution in terms of the actual contribution from 100 different intensity bins to mean precipitation. To account for the high frequency of low intensity precipitation events and the low frequency of high intensity events, we use an exponential bin distribution, as described by Berthou et al. (2019; see their Fig. S5). To calculate the contribution to mean precipitation, each bin frequency is multiplied by its average rate. Mean precipitation is therefore split in contributions of different rates. We consider a logarithmic scale on the x-axis, so the area under the curve is directly proportional to mean precipitation. Figure 1 shows the resulting distribution for PRIMAVERA, EUR-44 and observations over the Carpathian region (refer to Fig. S1 for the domain) in spring (MAM). Note that this type of histogram contains both information about precipitation mean (the area under the curve) and precipitation distribution. In this example, we see that PRIMAVERA simulates significantly less high-intensity precipitation than EUR-44 and is closer to observations. However, PRIMAVERA simulates too much low-intensity precipitation, a common bias among GCMs (Dai, 2006; Stephens et al., 2010). In the middle of the distribution, which represents moderate precipitation, the ensembles are not statistically different. These results are summarised in a pie plot (right panel of Fig. 1) for all seasons (DJF, MAM, JJA, SON).

The intercomparison of the model ensembles is performed as follows:

1) All datasets are regridded on the common EUR-44 rotated-pole grid, using a first-order conservative remapping. Then the precipitation data are pooled from each region and season. This step is repeated for every model and observational dataset.

2) On the histograms, the ensemble median is shown for each bin (thick line) along with the inter-quartile range (shaded colours in Fig. 1). For the observation, the median and inter-quartile range are based on inter-annual variability.

3) The significance of the difference between the two PRIMAVERA and CORDEX ensembles is calculated using a Welch's t-test (unequal variance t-test) for each bin based on inter-member spread (the $p$-value is plotted with grey crosses in Fig. 1). We apply a 10% threshold on each bin to test whether the two ensembles are significantly different ($p$-value $< 0.1$).

4) We group the bins in 3 intensity precipitation intervals defined on the observational datasets (Fig. 1): low (bins accounting for the lowest 40% of mean precipitation), moderate (bins accounting for the next 50% of mean precipitation: 40%-90%), high (bins accounting for highest 10% of mean precipitation). For each interval, we evaluate the percentage of bins over which the ensembles differ.

5) If the ensembles differ by more than 70% over that interval, the section of the pie corresponding to the season, region and precipitation interval is coloured (Fig. 1, right panel).

6) If the ensembles differ by more than 70%, we determine which one is less significantly different from the observations using the same metric between the observational spread (based on inter-annual variability) and the ensemble spreads

(based on inter-member spread). If the difference between an ensemble and the observations is at least 10% smaller than the other ensemble, its first letter is added to that section of the pie (P and C stand for PRIMAVERA and CORDEX, respectively). If the two ensembles are both close to observations (both differ by less than 30% with the observations), we add an "=" sign to the pie section.

These steps are performed for every season, region and precipitation intensity interval, and plotted as shown in Fig. 1. The pie plot is therefore a way to synthesize information for the comparison between CORDEX and PRIMAVERA (Sect. 3.3). It is worth noting that the results shown in the pie plots are not very sensitive to the size and distribution of the bins (not shown).

## 2.8 Sensitivity analyses

To evaluate the robustness of our results, we have performed several sensitivity analyses to assess:

- the role of model ensemble size by comparing the complete ensemble and a reduced ensemble
- the sensitivity of the results to the significance thresholds
- the robustness of the results when considering EUR-11 or EUR-44 RCMs
- the robustness of the results when considering observational uncertainties

These analyses are discussed in Sect. 3 and 4.

## 3 Results

### 3.1 Precipitation distribution in CMIP5, CORDEX and PRIMAVERA ensembles

Figures 2 and 3 show the precipitation distribution for PRIMAVERA, EUR-11, and a selection of CMIP5 models. The selection corresponds to the subset of GCMs that were downscaled by EURO-CORDEX RCMs: CNRM-CM5, CSIRO-Mk3-6-0, EC-EARTH, GFDL-ESM2G, GFDL-ESM2M, HadGEM2-ES, IPSL-CM5A-LR, IPSL-CM5A-MR, MIROC5, MPI-ESM-LR, NorESM1-M (refer to Table 2).

All data are regridded on the common EUR-44 grid but CMIP5 GCMs, which have been kept on their native coarse grids.

In Winter (Fig. 2), there is a clear shift in the precipitation distribution going from CMIP5 to PRIMAVERA and EUR-11 over all regions. PRIMAVERA and EUR-11 simulate an overall decrease in low intensity precipitation and an increase in high intensity precipitation compared to CMIP5. The shift towards more intense precipitation can be seen in all regions but is particularly clear over coastal and orographic regions (SC, AL, IP), which is presumably attributed to the finer horizontal grid spacing (Torma et al., 2015; Prein et al., 2016; Iles et al., 2019).

In Summer (Fig. 3), these findings are still valid between CMIP5, PRIMAVERA and CORDEX. CMIP5 simulates little high intensity precipitation, particularly over orographic regions, which shifts its distribution towards lower and moderate precipitation. PRIMAVERA and CORDEX simulate precipitation distributions closer to observations.

Both PRIMAVERA and CORDEX ensembles improve similarly upon CMIP5. This finding is attributed to their finer grid spacing (meaning the precipitation rates are those of a smaller area) and the better representation of orography and coastlines that enhance the triggering of precipitation. The effect of resolution is therefore the most important aspect to capture a realistic distribution of daily precipitation contribution to each precipitation rate. This finding is similar to Iles et al. (2019).

Analyses have also been performed on all CMIP5 GCMs available on ESGF (not shown). We have found that the ensemble mean (area under the curve) is slightly lower when considering all CMIP5 models, but the distribution does not shift, so our main conclusions do not change.

## 3.2 Mean differences between EURO-CORDEX and PRIMAVERA ensembles

Figure 4 shows mean biases compared to observations for the three ensembles: PRIMAVERA, EUR-44 and EUR-11. All ensembles overestimate winter precipitation (Fig. 4, top) over most European regions. The biases are reduced in PRIMAVERA over Western and Northern Europe but remain large in Eastern Europe. The results for spring (Fig. 4, second row) are also very similar, although PRIMAVERA overestimates precipitation in Scandinavia in this season. In summer, the ensembles are generally closer to observations, but they all show a dry bias in Eastern Europe and a wet bias along the Northern coastlines of Scandinavia -although observations likely underestimate rainfall in these mountainous regions (Lussana et al., 2018)-. EUR-11, and to a lesser extent EUR-44, overestimate precipitation in Western and Northern Europe in this season while PRIMAVERA is closer to observations. In autumn (Fig. 4, last row), precipitation is also overestimated in Northern and Central Europe in EUR-11 and EUR-44, while PRIMAVERA shows smaller biases. Over the Mediterranean coasts in autumn, all three ensembles underestimate precipitation over southeastern France and southern Alps. Berthou et al. (2020) and Fumière et al. (2020) showed that convection-permitting models are best to capture heavy precipitation events in these regions in autumn, which mostly contribute to mean precipitation. Note the sharp gradient in Italy between a large mean dry bias in the north and a large mean wet bias in the south. This reflects a problem in the observations, where E-OBS (covering the south) largely underestimates autumn precipitation in this region, which mostly falls as heavy precipitation (Flaounas et al. 2012).

In all seasons, EUR-11 and EUR-44 generally simulate more precipitation over orography than PRIMAVERA, showing large positive biases. However, precipitation observations are likely underestimated over orography (see discussion in Sect. 4.3). Overall, PRIMAVERA and EURO-CORDEX are best in summer and autumn. They suffer from large wet biases in winter and spring, although PRIMAVERA has a smaller bias. All biases shown still hold when considering the reduced ensembles, including only shared GCM families between PRIMAVERA and EURO-CORDEX driving GCMs (Fig. S2).

Figure 5 shows complementary information regarding the ability of PRIMAVERA and EURO-CORDEX (EUR-44 and EUR-11) to represent the spatial distribution of seasonal mean precipitation. The information is summarized as Taylor (2001) diagrams for all regions, seasons and ensembles, as compared to observations. Note that, even if Taylor diagrams are insensitive to mean biases, precipitation distributions are left bounded and standard deviations (distance to the origin in the

Taylor diagram) are related to mean precipitation (Casanueva et al., 2016a). The largest differences among the PRIMAVERA, EUR-44 and EUR-11 ensembles occur in regions with complex orography and land-sea contrasts (AL, CA, IP, MD, SC; connected symbols in Fig. 5). For the rest of the regions, the different ensembles perform similarly. There is a quite systematic behaviour across different regions, with PRIMAVERA closer to observations and EUR-44 showing an increasing error (both in terms of reduced spatial correlation and an overestimated spatial standard deviation). In winter, EUR-11 tends to overestimate even further the standard deviation (linked to the excessive mean precipitation shown in Fig. 4). For the rest of the seasons, EUR-11 tends to improve upon EUR-44, especially in terms of pattern correlation. EUR-11 reaches a correlation similar to PRIMAVERA, but still overestimates the observed spatial variability. This improvement in the spatial pattern of mean precipitation from EUR-44 to EUR-11 is in agreement with previous results (Casanueva et al. 2016b). Although EUR-44 uses a horizontal grid spacing that is similar to PRIMAVERA, its spatial distribution of precipitation is not as good as PRIMAVERA.

### 3.3 Daily precipitation distribution in CORDEX and PRIMAVERA ensembles

Figures 6 and 7 show daily precipitation distributions over different European regions in winter and summer, respectively. In both seasons, PRIMAVERA has significantly less heavy precipitation rates than EUR-11 in most regions, and is closer to observations (apart from BI in winter; Fig. 6). The reduced mean wet bias in PRIMAVERA mostly comes from less moderate and intense precipitation, especially in winter (Fig. 6). PRIMAVERA tends to have slightly more light precipitation than EUR-11, especially in winter. Note that the ensemble spread is generally larger in EUR-11 than PRIMAVERA, due to its larger ensemble, especially in summer when RCM simulations of precipitation are less constrained by GCM large-scale circulation. Over the Alps, PRIMAVERA underestimates summer heavy precipitation while EURO-CORDEX overestimates winter heavy precipitation (Fig. 2 and 3). Over the British Isles, heavy precipitation is underestimated by both ensembles in winter (Fig. 6), spring and autumn (not shown). In the Carpathians, summer precipitation is underestimated (Fig. 3) and winter precipitation is overestimated (Fig. 2) by both ensembles, although the biases are lower in PRIMAVERA. In Winter, both ensembles also largely overestimate precipitation in Central Europe, North East Europe and France (Fig. 6). Winter is the season with the largest precipitation undercatch in snow-dominated climates, so observations may be underestimated (Sect. 4.2).

Figure 8 gives an overview of significant differences between PRIMAVERA and EURO-CORDEX (EUR-11 and EUR-44) ensembles for each region, season and precipitation rate interval (low, moderate and heavy), by applying the method described in Sect. 2.7 and Fig. 1. In most regions and seasons, EURO-CORDEX and PRIMAVERA significantly differ from each other (the section of the pie is coloured) for the most intense precipitation rates. EUR-11 generally shows a heavier precipitation tail in all regions, which is often significantly larger than PRIMAVERA (e.g. in IP, CA, AL, MD, CE, FR regions; Fig. 2, 3, 6 and 7). This also applies to EUR-44, although to a lesser extent (Sect. 4.2). PRIMAVERA shows less contribution from these strong precipitation events, in better agreement with observations in most regions except over the Alps. In the Alps, the tail of the distribution indicates that heavy precipitation is overestimated by EURO-CORDEX and underestimated by PRIMAVERA

(Fig. 2 and 3), so observations lie in between the two ensembles. EUR-11 is closer to observations over the Alps in all seasons but winter, which appears to be too wet in EUR-11 (Fig. 2 and 4).

EUR-11 and PRIMAVERA do not differ significantly in light precipitation rates and only differ in moderate precipitation rates in FR and CE in spring and summer, and in BI in summer (Fig. 8a). This can seem counter-intuitive as mean biases are significantly closer to observations in PRIMAVERA (Fig. 4). Figures 6 and 7 actually show that, although PRIMAVERA is drier than EUR-11, it is often located within the ensemble spread of EUR-11. This indicates that some EUR-11 simulations perform similarly as PRIMAVERA for the moderate precipitation rates. This hypothesis is confirmed when the same GCM family is used (Fig. 8b). The differences between PRIMAVERA and EUR-11 in moderate precipitation intervals indeed disappear. However, they remain for high precipitation intervals in most regions and seasons.

PRIMAVERA are usually closer to observations in moderate and high precipitation (more "P" than "C" on Fig. 8a), but the ensembles are more similar when the same model family is considered (more "=" sign). This is also verified by building the same pie plot for each ensemble versus observations (Fig. S4). In light precipitation, both ensembles are not statistically different and are further away from observations. PRIMAVERA biases against observations are mostly significant in winter and spring. In winter they generally come from light and moderate precipitation and in spring they can come from different intervals depending on the region. EUR-11 is closer to observations mostly in the British Isles and the Alps.

In contrast to EUR-11, EUR-44 is generally closer to PRIMAVERA (Fig. 8c versus Fig. 8a), particularly in the most intense precipitation, where EUR-11 shows a large increase in intense precipitation compared to EUR-44 (the differences between EUR-44 and EUR-11 are discussed further in Sect. 4.3). As for EUR-11, reducing the EUR-44 ensemble to the models which are common to each ensemble also tends to make the ensembles less statistically different, and closer to observations (Fig. 8d).

## 4 Evaluation of the robustness of results

### 4.1 Sensitivity of results to significance thresholds

Our results are based on model ensembles. Different conclusions may be drawn either when evaluating models individually (e.g. Klingaman et al., 2017, show large differences in the character of rainfall in different models) or with a slightly different selection of models within the ensembles. To evaluate the ensemble spread, we have used an interquartile range. For comparison, we have also used a bootstrap resampling method applied 1000 times by selecting different models in each ensemble, and have found similar results (not shown).

To determine whether the results shown in the pie plots depend on the chosen significant threshold (5% significance level on 70% of the interval), we have performed sensitivity analyses on: 1) the threshold that defines the level of significance ($p$-value of 10 or 5%); 2) the percentage of the interval on which bins are significantly different (70 or 90%). The results are summarised in Fig. S8. When strengthening both the significance level (5%) and the percentage over the interval (90%) (top right panel),

the ensembles are hardly different in regions that are located at the boundaries of the EURO-CORDEX domain (IP, NEE, CA, MD, SC) but remain different in Central Europe (BI, CE, FR, AL). If only one of the criteria is strengthened (top left and bottom right panels), differences in the highest precipitation interval remain in Central and Western Europe in winter and spring, and in Northern and Eastern Europe in summer.

When loosening the criteria defining when both ensembles are close to observations ("=" signs in the top right panel), it is clear that both ensembles are best at capturing summer and autumn precipitation in most regions (except SC and IP).

## 4.2 Sensitivity of results to the choice of EUR-44 or EUR-11

We have focused most of the analyses on EUR-11 which represents a larger ensemble than EUR-44, based on state-of-the-art RCMs, while EUR-44 RCMs are slightly older. EUR-11 also has a finer grid spacing which is important in regions with complex topography (orographic and coastal regions), especially in winter (Fig. 2). The horizontal grid spacing used in EUR-44 is on the other hand more similar to PRIMAVERA so EUR-44 is included in the analyses to discuss the impact of resolution in EURO-CORDEX ensembles on daily precipitation distribution.

Figure 4 shows that EUR-11 precipitation mean biases are similar or larger than EUR-44. EUR-11 is generally wetter. This finding is in line with previous studies showing no systematic improvement between EUR-44 and EUR-11 for mean precipitation (Kotlarski et al., 2014; Casanueva et al., 2016b; Iles et al., 2019). However, Fig. 5 shows that the spatial correlation of mean precipitation is improved in EUR-11 compared to EUR-44 in all seasons but in winter, and in all regions particularly over orography (AL). This is attributed to EUR-11's finer grid spacing. These results are very similar to previous studies that assessed the added value of EUR-11 over EUR-44 (Kotlarski et al., 2014; Torma et al., 2015; Prein et al., 2016). The results are also similar to previous studies regarding the tail of precipitation distribution. EUR-11 generally simulates more high intense precipitation, particularly over orography (Torma et al., 2015; Prein et al., 2016; Iles et al., 2019; Fig. S5 and S6). EUR-44 is therefore generally closer to PRIMAVERA than EUR-11 for the heavy precipitation interval, particularly in winter. This is attributed to the coarser grid of EUR-44, which is similar to PRIMAVERA.

Overall, as shown by Fig. 8, the conclusions found against PRIMAVERA are valid whether we consider EUR-11 or EUR-44. These analyses have been performed on the common EUR-44 grid. Leaving the EUR-11 ensemble on its native grid tends to shift the distribution towards higher intensity precipitation over most regions, particularly over orography and coastal regions (not shown). This has been shown by Torma et al. (2015) over the Alps. Similarly, remapping the high-resolution gridded observations on the EUR-11 grid tends to slightly change the distribution over most regions and seasons, but remains within the observational interannual variability (not shown).

## 4.3 Observational uncertainty

Prein and Gobiet (2017) advised to consider as many observational datasets as possible for regional analyses. Most datasets, however, are available either at much coarser horizontal grid spacing than 50 km and therefore cannot be used for evaluating the ensembles at such resolution, or they are not available at daily timescales. We have done a test using GPCP v2 available daily at 1 degree horizontal resolution. The distribution shows almost no intense precipitation over most regions and in most seasons (not shown). The observational datasets used in this study are available at fine horizontal grid spacings (5-20 km) and they contain a very dense stations network, which minimizes the effect of precipitation undersampling (Prein and Gobiet, 2017). These data are therefore considered to be the best available over Europe. Nevertheless, there are drawbacks, particularly related to the lack of precipitation undercatch correction. Precipitation undercatch strongly depends on precipitation phase, intensity, location of stations (i.e. their exposure), and the type of gauge used (shielded vs. unshielded). This issue can be particularly important for falling snow over mountains but also in other places when associated with strong winds (in that case, rain does not fall vertically in the gauges, which creates an error depending on wind speed and drop size). The lack of correction can include errors of 3-20% on average and up to 40-80% in high latitudes and mountainous regions (Prein and Gobiet, 2017). Besides precipitation undercatch, another error source that affects specifically heavy rainfall in gridded observations is the smoothing of intense rainfall over larger regions. This occurs in all gridded precipitation datasets that are purely based on gauges and results in dampening extreme precipitation peaks by smoothing the heavy rainfall and redistributing it over larger regions (Haylock et al., 2008; Hofstra et al., 2009; Prein and Gobiet, 2017). To represent the spread in observations, we use the inter-annual variability of the gridded observation datasets. However, the heaviest precipitation interval is likely underestimated in our analyses. To overcome this problem, we redid some of our analyses by assuming a mean estimate of the undercatch error of 20% over all regions and seasons, a method similar to Kotlarski et al. (2014) and Rajczak and Schär (2017). In such a method, all observations are scaled by a factor of 1.2 over all grid points and over the entire time series, which gives a rough estimate of observational uncertainties. In such a case, our analyses become almost systematically more favourable to both EUR-11 and EUR-44, which simulate more heavy precipitation than PRIMAVERA (Fig. S7). In the MD region where E-OBS is used, both PRIMAVERA and CORDEX simulate much more intense precipitation than observations in winter and autumn (Fig. 6 and S3). This is likely due to the small rain gauge density in E-OBS in this region and a large underestimation of heavy rainfall (Flaounas et al., 2012). Therefore, comparisons with observations in this region should be taken cautiously.

## 5 Summary and Discussion

### 5.1 Summary

In this study, we have considered high-resolution PRIMAVERA GCMs of HighResMIP (25-50 km horizontal grid spacing) and EURO-CORDEX RCMs (12-50 km horizontal grid spacing) present-day simulations to evaluate the ability of these

ensembles to represent daily precipitation distribution over Europe. This study is the first attempt to evaluate GCM and RCM ensembles provided at similar horizontal resolutions at the regional scale.

Our results show that CMIP5-driven EUR-44 and EUR-11 RCMs, and PRIMAVERA atmosphere-ocean coupled GCM ensembles give equivalent regional climate information in terms of daily precipitation distribution and its contribution to precipitation intervals at a horizontal grid spacing of 50 km. The differences in their precipitation distribution are smaller than differences between CORDEX and CMIP5, where the value of higher resolution models is indisputable (Fig. 2 and 3). CMIP5 models show rather different distributions, particularly shifted to lower precipitation intensities, as expected from their coarse resolution (Iles et al., 2019). This added value of CORDEX RCMs to CMIP5 GCMs emphasizes the importance of a well designed, well evaluated model chain when using dynamical downscaling as a method to obtain higher resolution climate data. PRIMAVERA and CORDEX ensembles are of good quality in summer and autumn but tend to overestimate precipitation in winter and spring. This bias is reduced in the PRIMAVERA ensemble. Regarding EUR-11 and EUR-44, this wet bias was also identified for older evaluation simulations (Kotlarski et al. 2014), suggesting that this is a local bias, common to many RCMs. The reduction of the EURO-CORDEX ensemble to the same GCM family as used in PRIMAVERA does not reduce this bias, suggesting that this bias is essentially due to the RCM, and not caused by the large-scale biased circulation of the GCM.

There are some precipitation intervals, seasons and regions for which the two ensembles significantly differ. A major difference between the two ensembles is found for heavy precipitation in most seasons and regions. PRIMAVERA has less heavy precipitation than EURO-CORDEX, and tends to be closer to observations. However, gridded observational datasets likely suffer from an underestimation of heavy precipitation, in which case EURO-CORDEX is favorable (Fig. S7). European summer precipitation is mostly driven by local convective precipitation, which is not explicitly simulated in state-of-the-art EURO-CORDEX RCMs and GCMs. At such resolutions (at best 12 km horizontal grid spacing), convection is parameterized. In RCMs, such parameters are commonly set by expert tuning or objective calibration to simulate a mean climate as close as possible to observations over the region of interest in hindcast simulations (using reanalysis boundary forcings; e.g. Bellprat et al., 2016). It is not possible to perform such tuning in GCMs. GCMs are commonly tuned to balance top-of-the-atmosphere radiation globally or to better represent specific processes, but cannot be tuned over a specific region (Hourdin et al., 2017). Another hypothesis to explain this excess in precipitation in the CORDEX ensemble is that RCMs do not often use the semi-implicit semi-Lagrangian numerics commonly used in GCMs that allow for longer time steps. Using shorter time steps tends to increase both mean and extreme precipitation, while long semi-implicit time steps appear to smooth the results (Zeman et al., in prep.). PRIMAVERA tends to have more light precipitation than EURO-CORDEX, and too much compared to observations, although this result is not as robust as the former one. It is possible that the selection of the convective scheme and land-surface scheme in RCMs has a positive effect towards reducing this "drizzling" problem.

When only considering shared GCM families between the CORDEX and PRIMAVERA ensembles, differences in the bulk of the distribution (low and medium precipitation rates) vanish in almost all regions and seasons, and particularly in winter (Fig.

8). This suggests an important role of the driving model in the quality of the RCM simulations. However, mean biases remain in EURO-CORDEX and are still larger than in PRIMAVERA (Fig. S2). PRIMAVERA is a small ensemble, and its results are mostly within the range of the large EURO-CORDEX ensemble (particularly EUR-11; Fig. 6 and 7). This suggests that a careful choice of a subset of EUR-11 may perform as well as PRIMAVERA.

## 5.2 Discussion

The performance of PRIMAVERA was not logically expected. If we were to compare atmospheric-only simulations, as in Denis et al. (2002) and Iles et al. (2019), the forcing imposed by observed sea surface temperatures would drive the GCMs and RCMs towards more realistic, and therefore potentially more similar, responses at the regional scale. This could be verified by comparing the high-resolution HighResMIP/PRIMAVERA atmospheric-only simulations with the EURO-CORDEX evaluation simulations driven by reanalyses. Here, PRIMAVERA and EURO-CORDEX historical simulations are coupled GCMs and coupled GCM downscaling, respectively. Including ocean coupling inevitably includes differences in large-scale circulation between low-resolution (CMIP5) and high-resolution (PRIMAVERA) GCMs, which result in different simulations of the regional climate. Moreover, although high-resolution GCMs have the potential to better simulate large-scale circulation, which should improve the regional climate (Roberts et al., 2018; Gutowski et al., 2020), in the case of HighResMIP, the GCMs are not tuned for higher resolution (see Roberts et al., in revision, for changes in models when increasing resolution), and the experimental design is rather simplified (e.g. simple aerosol, short spinup, etc.; Haarsma et al., 2016). In contrary, although RCMs downscale low-resolution coupled GCMs and so inherit their biases in terms of large-scale circulation (Gutowski et al., 2020), RCMs have the main advantage of being tuned over the region of interest, and often correct the GCM biases (e.g. Sørland et al., 2018). We show here that once GCMs use competitive resolution, they produce reasonable results, even with some simplifications from the experimental design.

The fact that PRIMAVERA results exhibit moderate improvements over CMIP5-driven CORDEX simulations in some regions and seasons with our precipitation metrics is consistent with the results of Iles et al. (2019) who used a very different method to compare GCMs and RCMs at different horizontal resolutions. It indicates that the potential improvement of large-scale dynamics in high-resolution GCMs has a positive influence on precipitation distribution.

This study is a first effort to evaluate the quality of regional climate information provided by GCM and RCM ensembles of similar horizontal grid spacings. We have only investigated daily precipitation distribution, and such an exercise needs to be continued with other fields (temperature, winds) for mean, variability and extremes. Nevertheless, the results are very promising, in particular as the two ensembles have similar performances when compared on a common grid spacing of 50 km. PRIMAVERA and EURO-CORDEX (EUR-11 or EUR-44) should therefore be considered equally credible, depending on the

user's needs. For studies at the local scale or over complex orography (such as the Alps), a higher resolution model dataset, such as EUR-11, gives more detailed spatial information (e.g. Kotlarski et al., 2014; Prein et al., 2015).

We have only focused on present-day simulations. Assessing future climate projections between the two ensembles may be more difficult because the results would depend on other parameters independent of the models themselves, such as the lack of a common protocol (e.g greenhouse gases and aerosols forcings) between RCMs and GCMs. The impact of aerosol forcings on the climate projections, which differ in GCMs and RCMs, is currently being investigated (Boé et al., 2020; Gutierrez et al., 2020).

We have limited our study to Europe, which has the advantage of having a large RCM ensemble. We showed that GCMs have the potential of providing, at global scale, regional climate information which is on a par with CORDEX datasets. Therefore, in the future, this work should be extended to other regions of the world, where CORDEX-22 and CORDEX-44 ensembles can be compared to HighResMIP GCMs. In this respect, CORDEX-22 simulations following the CORE protocol (https://www.cordex.org/experiment-guidelines/cordex-core) will be especially valuable, since they provide a core set of comprehensive and homogeneous regional climate projections across many domains of the globe (Gutowski et al., 2016), which can be compared to the global high-resolution information provided by HighResMIP (e.g. Hariadi et al., in prep).

**Opportunities for future coordination**

By design, RCMs will always be able to run at a smaller grid spacing than GCMs for a given computational power. However, for the first time in the history of climate modelling, an ensemble of cutting-edge GCMs has reached a grid spacing comparable to that of standard RCM ensembles. This was a major effort by the GCM community, which brings a new opportunity for collaboration with the RCM community. From this point on, both communities will provide complementary results at each resolution increase, fostered by an ever increasing computational power. In the same vein, the RCM community efforts are now directed towards kilometer-scale, convection-resolving climate modeling, which has shown promising results particularly regarding the representation of heavy short-term precipitation and in reducing modeling uncertainty (Prein et al., 2013; Ban et al., 2015; Prein et al., 2015; Giorgi et al., 2016; Schär et al., 2020, Berthou et al., 2020). With this effort, the RCM community has reached the grid spacing of limited-area mesoscale modeling, in use for decades by the Numerical Weather Prediction (NWP) community for process understanding and case study analyses (Coppola et al., 2020). In parallel to the resolution increase, both GCM and RCM communities are increasing the complexity of their models, incorporating new components of the Earth system and new processes within them. Therefore, the numerical weather and climate modelling community, as a whole, is currently at a turning point where its results can, and will, be compared. Although the coordination of GCM and RCM simulations has made rapid progress recently (e.g. CMIP, HighResMIP, EURO-CORDEX (through C3S-PRINCIPLES), CORDEX-CORE, CORDEX Flagship Pilot Simulations), the increasing resolution and complexity of both GCMs and RCMs will advance the need for enhanced coordination to produce results that can be fully compared, especially regarding climate change projections (e.g. Boé et al., 2020). In particular, there is still very limited coordination of km-resolution scenario simulations, and a joint and coordinated database (such as available for CMIP6 and CORDEX) is still missing. Convection-

resolving simulations can now be run at decadal scale, but they are still too expensive to provide multi-model ensembles of centennial climate change projections. End users therefore have to rely on a combination of sources (CMIP, conventional and km-resolution CORDEX) for adaptation purposes. Similarly, our results regarding precipitation distribution show that high-resolution GCM and CORDEX simulations could be combined in a joint archive, following the respective format and data standards, thereby making it more convenient for impact groups to use these simulations.

## 6 Code availability

The code used for the analyses presented in this manuscript is developed by Berthou et al. (2019) available under the terms of the Apache 2.0 license from https://github.com/PRIMAVERA-H2020/PrecipDistribution (DOI: 10.5281/zenodo.3956780). This code uses the method that computes precipitation histograms of the contributions of specific intensity bins to the total precipitation based on the ASoP1 diagnostics developed by Klingaman et al. (2017) and available under the terms of the Apache 2.0 license from https://github.com/nick-klingaman/ASoP.

## 7 Data availability

All PRIMAVERA and CORDEX model data used in this study can be obtained from the Earth System Grid Federation nodes, such as esgf-data.dkrz.de, esgf-index1.ceda.ac.uk, cordexesg.dmi.dk, esgf-node.ipsl.fr, and esg-dn1.nsc.liu.se. Note that the simulations are still being produced, so the ensembles presented in this article may not cover the whole ensembles available on the ESGF archive. For a complete list of EURO-CORDEX simulations, please refer to the EURO-CORDEX homepage (www.euro-cordex.net). Details about PRIMAVERA data availability can be found in Roberts et al. (2017), Roberts (2018), Scoccimarro et al. (2018), von Storch et al. (2018), EC-Earth Consortium (2018), Voldoire (2019). PRIMAVERA Persistent Identifiers (PID) and CORDEX file tracking IDs are listed in the Data folder of https://github.com/PRIMAVERA-H2020/PrecipDistribution (DOI: 10.5281/zenodo.3956780).

Spain02 are available at http://www.meteo.unican.es/datasets/spain02; PT02 are available at http://www.ipma.pt/pt/produtoseservicos/index.jsp?page=dataset.pt02.xml. EURO4M-APGD are available at https://www.meteoswiss.admin.ch/home/search.subpage.html/en/data/products/2015/alpine-precipitation.html (doi: 10.18751/Climate/Griddata/APGD/1.0). CARPATCLIM are available at http://surfobs.climate.copernicus.eu/dataaccess/access_carpatclim.php. The E-OBS data are obtained through the ECA&D project: https://www.ecad.eu.

*Author contributions*. The authors list is written by contribution (from M.-E. Demory to C. Schär), then in alphabetic order. M.-E. Demory and S. Berthou ran the analyses on all models and observations used in this study, based on the diagnostics developed by S. Berthou and previously published. S. Sørland and M. Roberts contributed to CORDEX and PRIMAVERA

analyses, respectively. J. Fernandez and R. Brogli strongly contributed to the revisions of the manuscript. U. Beyerle provided all CORDEX and CMIP5 data downloaded from ESGF. J. Seddon provided the PRIMAVERA data on CEDA JASMIN (Lawrence et al., 2013). R. Haarsma and C. Schär were strongly involved in the discussion of the results. The other co-authors contributed to run the simulations. M-E Demory wrote the manuscript, together with S. Berthou, J. Fernandez and S. Sørland, and with input from all other co-authors.

*Acknowledgement*. This work is funded through the EU Copernicus Climate Change Service (C3S): Producing Regional Climate Projections Leading to European Services (PRINCIPLES). The PRIMAVERA project is funded by the European Union's Horizon 2020 programme, Grant Agreement no. 641727. We acknowledge the World Climate Research Programme's Working Group on Regional Climate, and the Working Group on Coupled Modelling, former coordinating body of CORDEX and responsible panel for CMIP5. We also thank all the climate modelling groups (listed in Tables 1 and 2 of this paper) for producing and making available their model output. We also acknowledge the Earth System Grid Federation infrastructure, an international effort led by the U.S. Department of Energy's Program for Climate Model Diagnosis and Intercomparison, the European Network for Earth System Modelling and other partners in the Global Organisation for Earth System Science Portals (GO-ESSP).

ETH authors acknowledge PRACE for awarding us access to Piz Daint at ETH Zürich/CSCS (Switzerland) for conducting CCLM simulations. This work used JASMIN, the UK's collaborative data analysis environment (http://jasmin.ac.uk). S.B. gratefully acknowledges funding from the European Union under Horizon 2020 project European Climate Prediction System (EUCP; Grant agreement: 776613). J.F. acknowledges support from the Spanish R&D Program through project INSIGNIA (CGL2016-79210-R), co-funded by the European Regional Development Fund (ERDF/FEDER).

We acknowledge the E-OBS dataset from the EU-FP6 project UERRA (http://www.uerra.eu) and the data providers in the ECA&D project (https://www.ecad.eu). We acknowledge the CARPATCLIM Database © European Commission - JRC, 2013. The authors thank IPMA for the PT02 precipitation dataset, as well as AEMET and UC for the Spain02 dataset, available at http://www.meteo.unican.es/datasets/spain02. The SAFRAN dataset was provided by METEO FRANCE. The European Climate Prediction system, which provided UKCPobs, is funded by the European Union's Horizon 2020 programme, Grant Agreement no. 776613. We thank the Federal Office of Meteorology and Climatology MeteoSwiss for providing the Alpine precipitation grid dataset (EURO4M-APGD) developed as part of the EU project EURO4M (www.euro4m.eu).

The authors would like to thank the two anonymous referees for their thorough review and constructive comments that contributed to the improvement of this manuscript.

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

| Model name | HadGEM3-GC31-HM | EC-Earth3P-HR | CNRM-CM6-1-HR | MPI-ESM1-2-XR | CMCC-CM2-VHR4 | ECMWF-IFS-HR |
|---|---|---|---|---|---|---|
| Institute | Met Office | KNMI, SMHI, BSC, CNR | CERFACS | MPI-M | CMCC | ECMWF |
| Reference | Roberts et al., 2019 | Haarsma et al., 2020 | Voldoire et al. 2019 | Gutjahr et al., 2019 | Cherchi et al., 2019 | Roberts et al., 2018a |
| Atmosphere horizontal resolution (in km at 50N) | N512 (25km) | TI511 (36km) | TI359 (50km) | T255 (34km) | 0.25° (18km) | Tco399 (25km, output at 50km) |
| Ocean resolution (km) | 25km | 25km | 25km | 40km | 25km | 25km |
| Simulation | hist-1950 | hist-1950 | hist-1950 | hist-1950 | hist-1950 | hist-1950 |
| Ensemble member | r1i1p1f1 | r1i1p2f1 | r1i1p1f2 | r1i1p1f1 | r1i1p1f1 | r1i1p1f1 |

**Table 1: Information about the PRIMAVERA high-resolution GCMs used in this study, including their spatial resolution (for full details, refer to https://www.primavera-h2020.eu/modelling/our-models/). The ones listed in bold are of the same family as the CMIP5 GCMs downscaled by CORDEX.**

**Table 2: Summary of historical EURO-CORDEX simulations used in this study. The first column indicates HighResMIP models of the same family as the CMIP5 GCM (second column) driving the RCMs. Matching colors show comparable HighResMIP GCMs and EURO-CORDEX RCMs. Within EURO-CORDEX RCMs, dark shaded models are available at both 0.11º (EUR-11) and 0.44 (EUR-44) horizontal resolutions. HIRHAM5\* indicates several versions of this model were used. See Table S1 for the full list of EURO-CORDEX data used, including institutions and detailed RCM model version.**

| Observations | SAFRAN | UKCPobs | CARPATCLIM | SPAIN02 v2 + PT02 v2 | ALPS-EURO4M | E-OBS v17 |
|---|---|---|---|---|---|---|
| Domain covered | France | British Isles | Carpathians | Iberian Peninsula | Alps | Other European regions |
| Spatial resolution | 8km | 5km | 0.1° | 0.2° | 5km | 0.5° |
| Temporal resolution | daily | daily | daily | daily | daily | daily |
| Time period considered | 1971-2005 | 1971-2005 | 1971-2005 | 1971-2003 | 1971-2005 | 1971-2005 |

**Table 3: Information about the observational datasets used in this study (refer to Fig. S1 for the coverage). The time period concerns that considered in this study, not the available period of each observational datasets.**

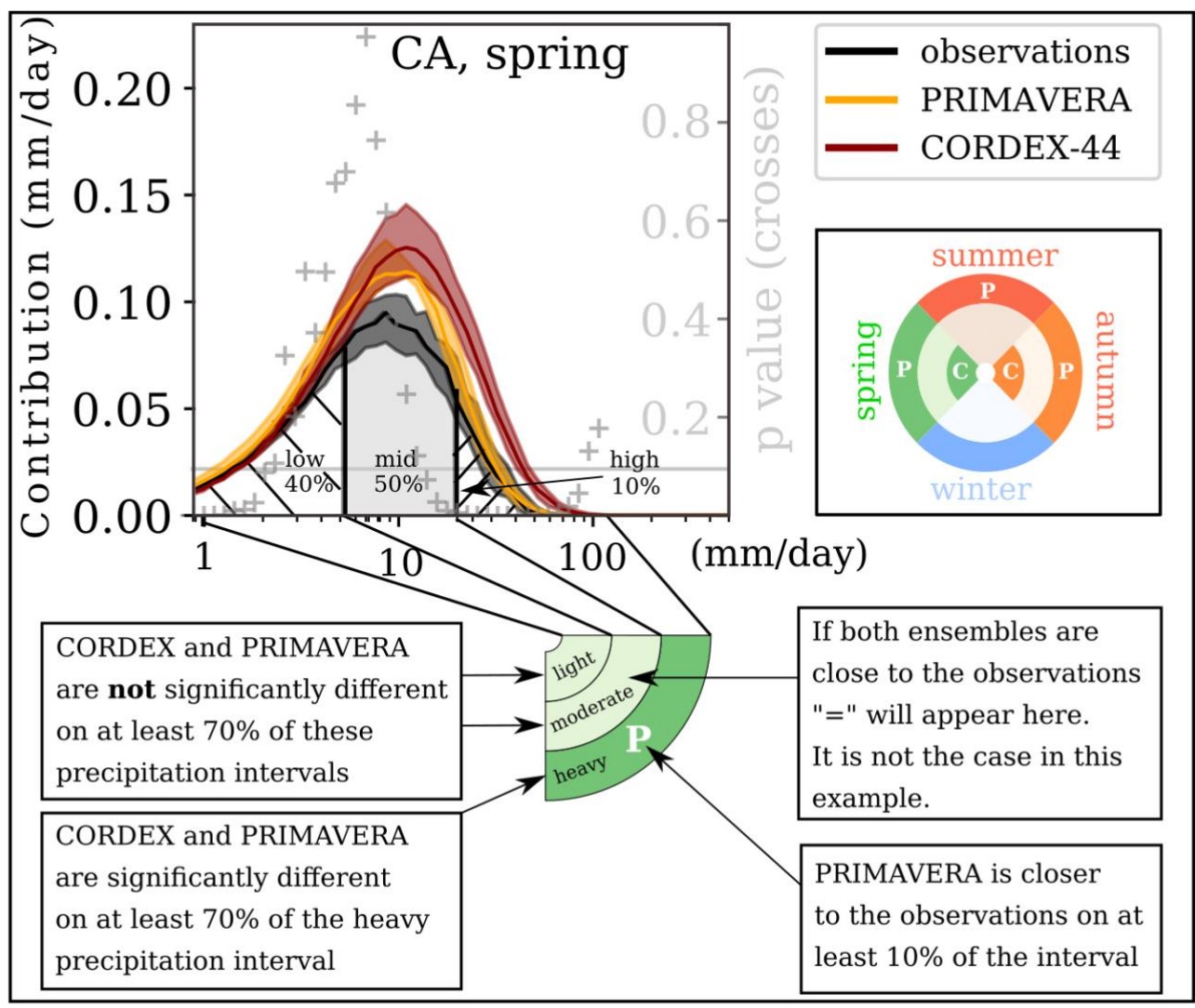

**Figure 1: Explanation of the method:(top)** Daily precipitation contribution to mean precipitation (precipitation frequency x bin intensity) using exponential bins in the Carpathian region (CA) in spring (MAM) for PRIMAVERA, EUR-44 and observations. The thick lines show the ensemble median, the model spread is calculated on the inter-quartile range of the inter-member spread, and the observation spread is based on inter-annual values. Grey crosses are *p*-values of the PRIMAVERA versus CORDEX difference using a Student's t-test, the ensembles are significantly different where the crosses are below 0.1. **(bottom)** For three precipitation intensity intervals (light: accounting for lowest 40% of mean precipitation, moderate: accounting for the next 50% of the mean, heavy: accounting for highest 10% of the mean), a pie is coloured if the ensembles differ on more than 70% of the interval. Letters show which ensemble is closest to observations ('P' for PRIMAVERA, 'C' for CORDEX; no colour indicates that none of the ensembles are close to observations, '=' indicates that both ensembles are close to observations). **(right)** Resulting pie for each region (here CA) and season.

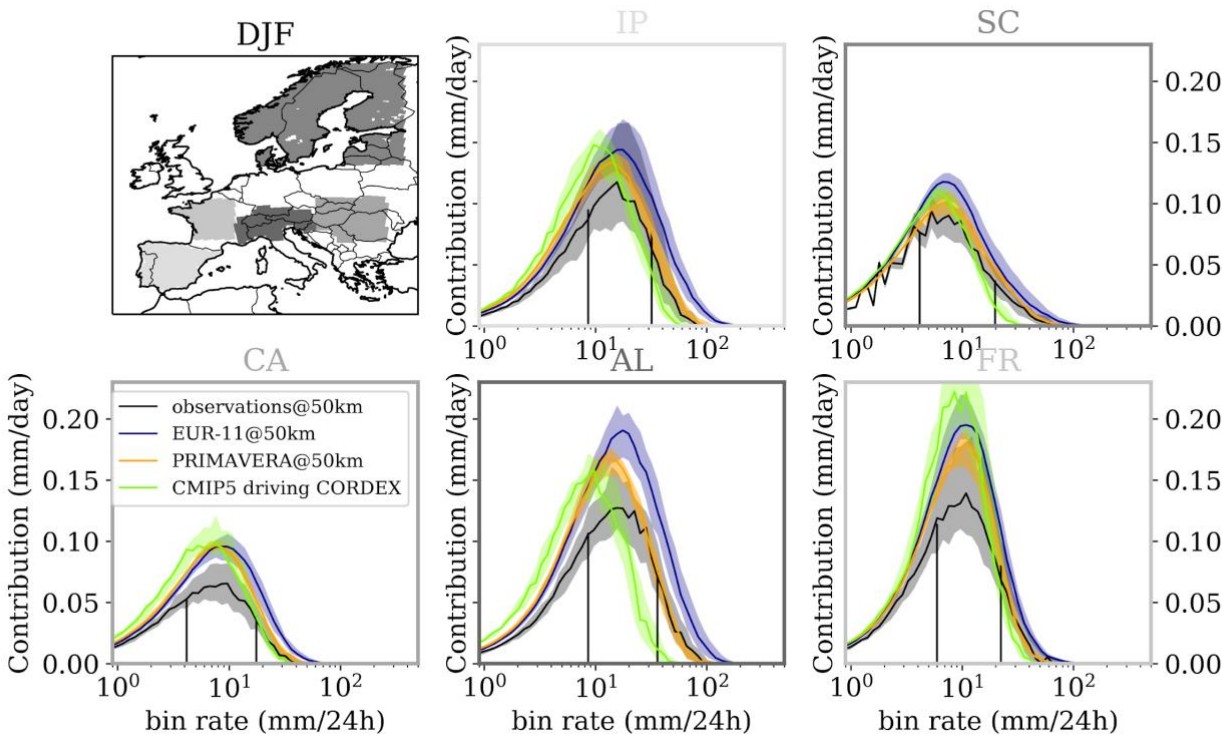

**Figure 2: Precipitation contribution (frequency x bin rate) per precipitation rate in DJF over the Iberian Peninsula (IP), Scandinavia (SC), the Carpathian region (CA), the Alps (AL), and France (FR), for a selection of CMIP5 GCMs (green), PRIMAVERA (orange), EUR-11 (blue). CMIP5 data are plotted on the models native grid, the other datasets are regridded on a common EUR-44 grid.**

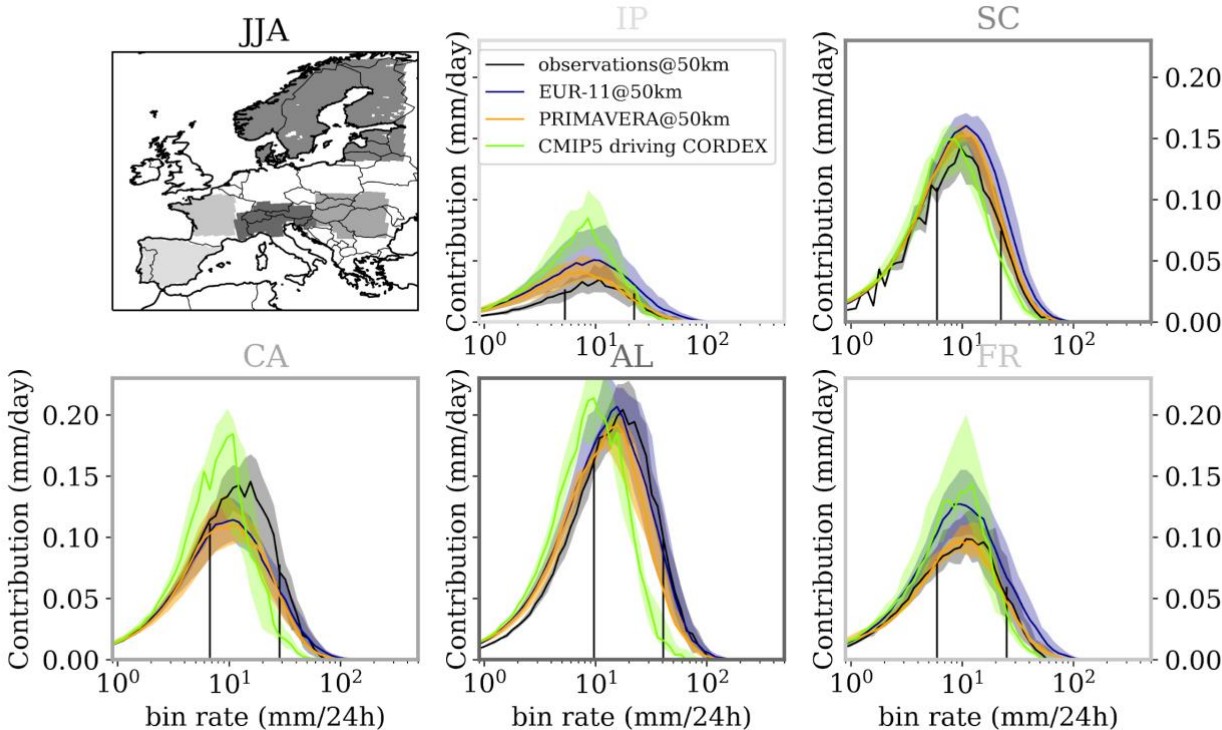

**Figure 3: As Fig. 2 for summer (JJA).**

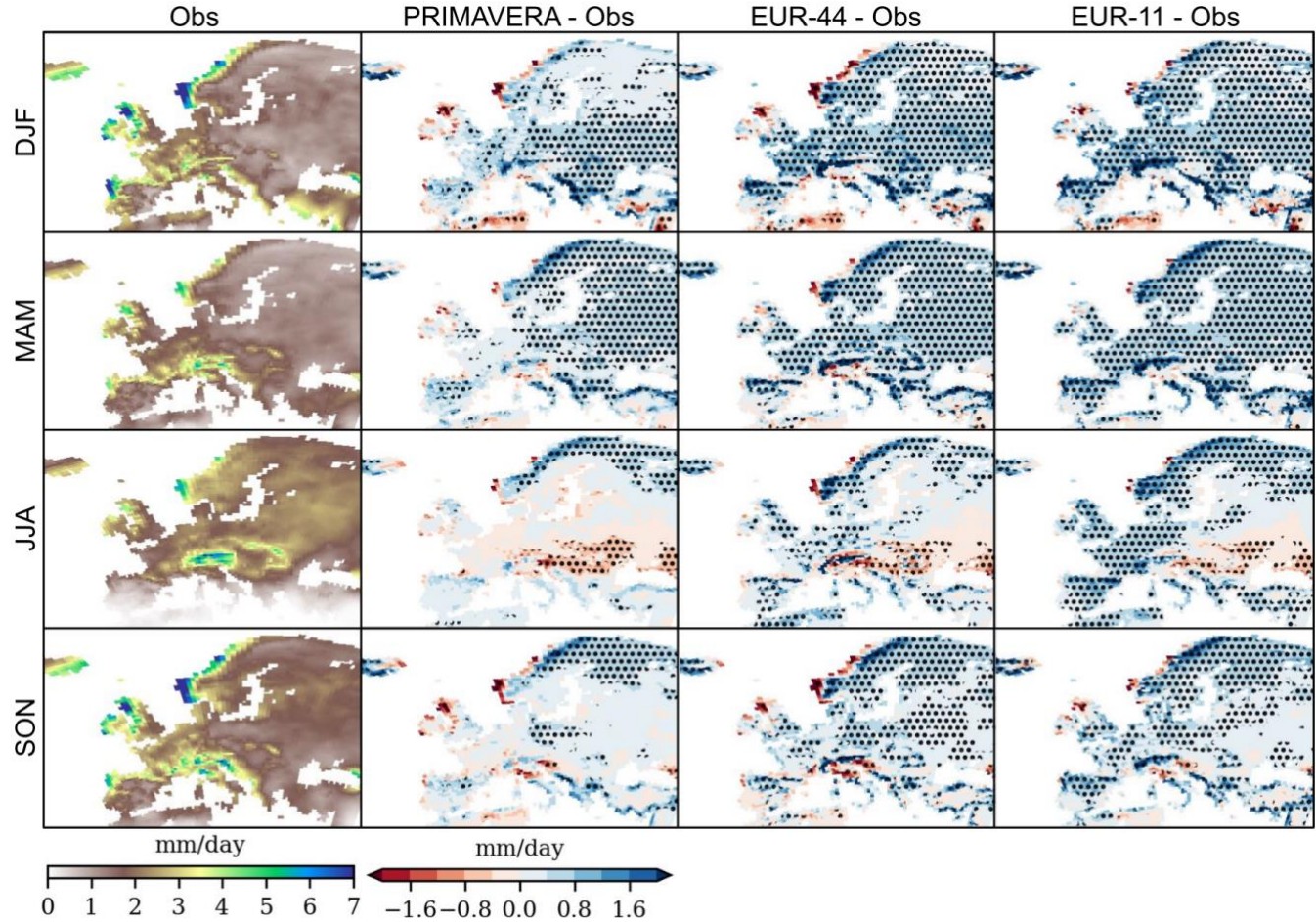

**Figure 4: Mean precipitation from the combination of observational datasets (first column), and mean precipitation bias between PRIMAVERA and observations (second column), EUR-44 and observations (third column), and EUR-11 and observations (last column). Results for the different seasons are shown in rows. Dots show regions were mean bias is statistically significant at the 10% level. All datasets are shown on the EUR-44 grid.**

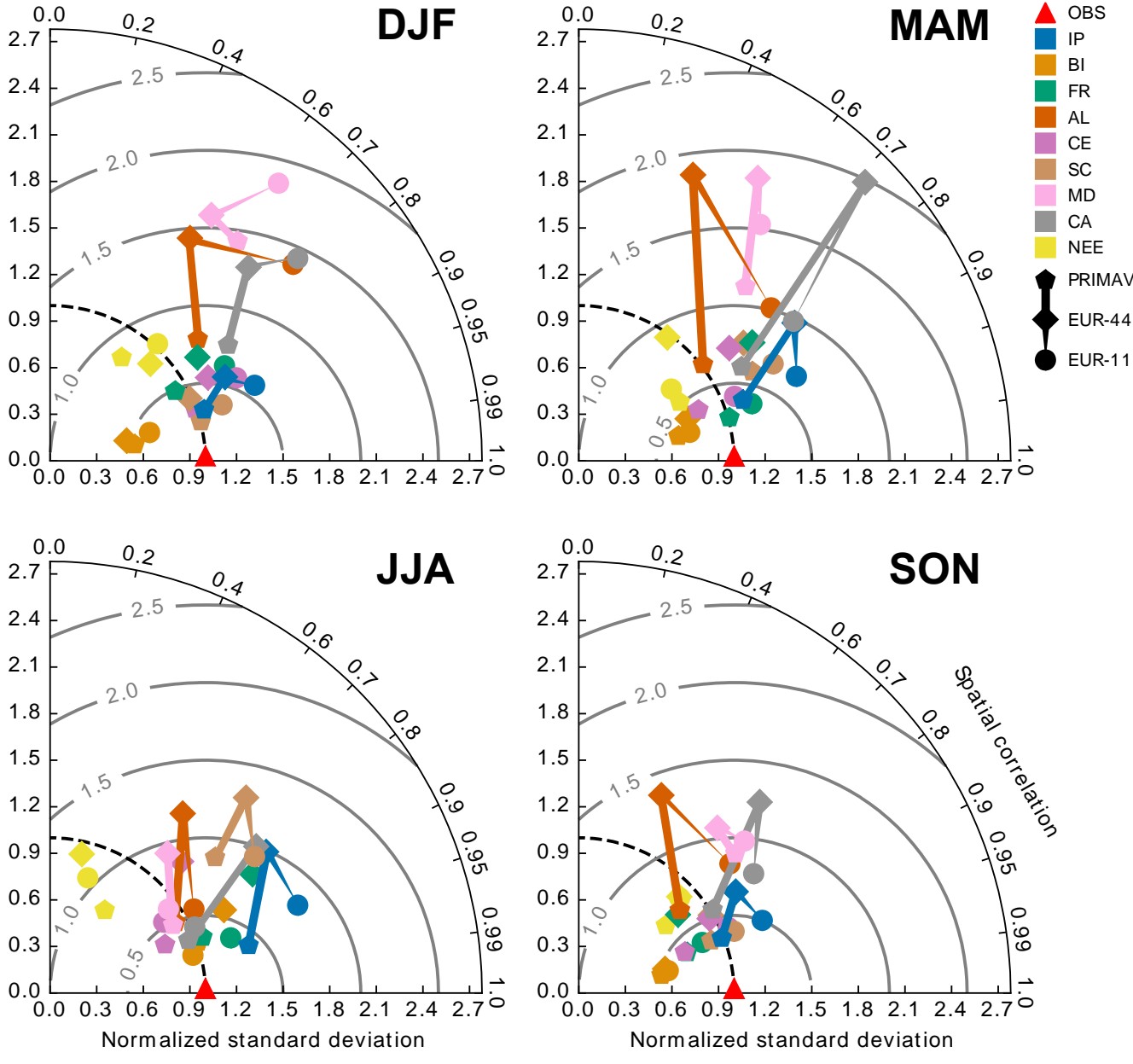

**Figure 5: Taylor diagrams performed on the spatial distributions of seasonal mean precipitation for EUR-11 (circles), EUR-44 (diamonds) and PRIMAVERA (pentagons) ensemble means for DJF (top left), MAM (top right), JJA (bottom left), SON (bottom right) over all regions. Symbols are connected (see legend) for complex-orography and coastal regions (AL, CA, IP, MD, SC). Observational references are shown in red triangles.**

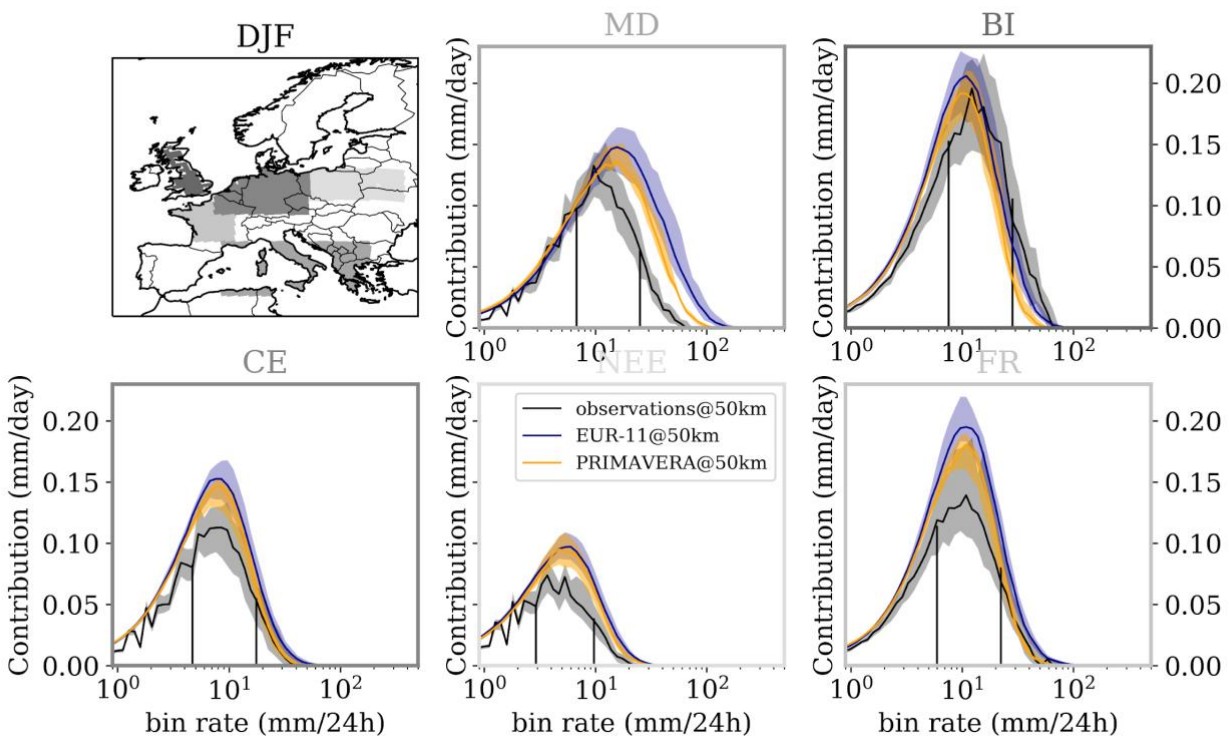

**Figure 6: Precipitation contribution (frequency x bin rate) per precipitation rate in DJF over the Mediterranean (MD), British Isles (BI), Central Europe (CE), North East Europe (NEE), France (FR) for EUR-11 (blue), PRIMAVERA (orange), and observations (black) regridded on EUR-44.**

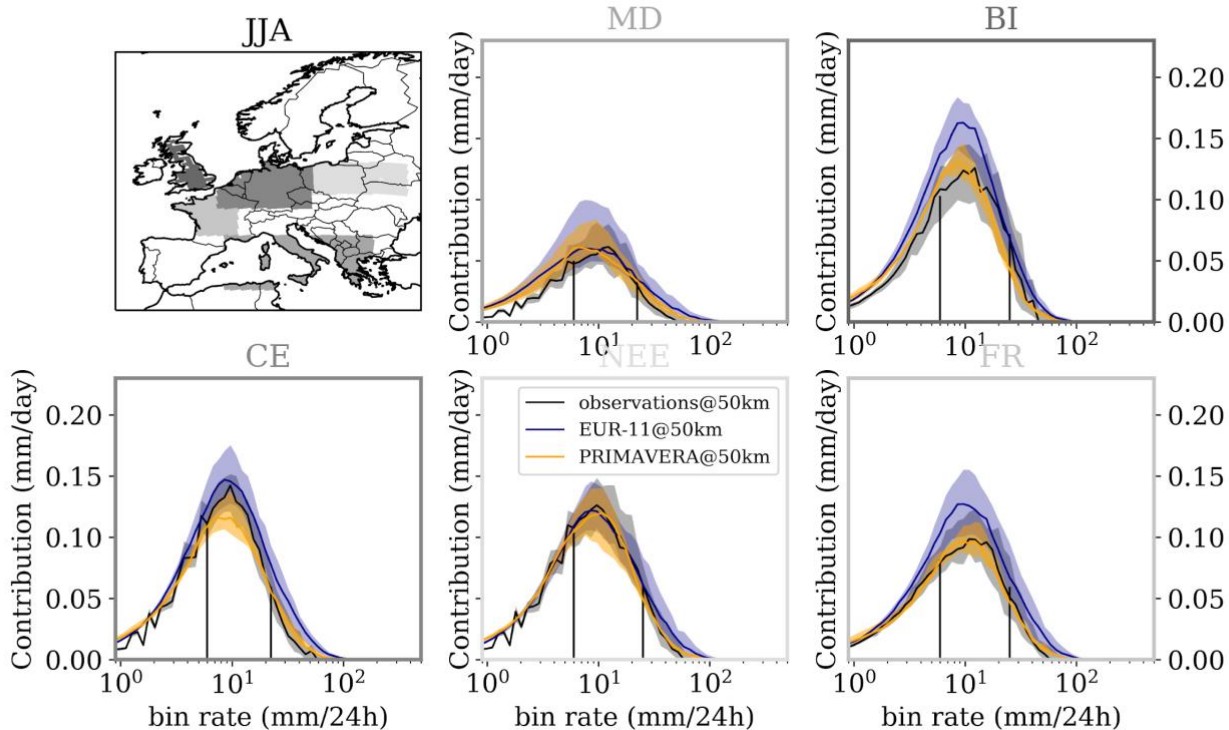

**Figure 7: Same as Fig. 6 for JJA.**

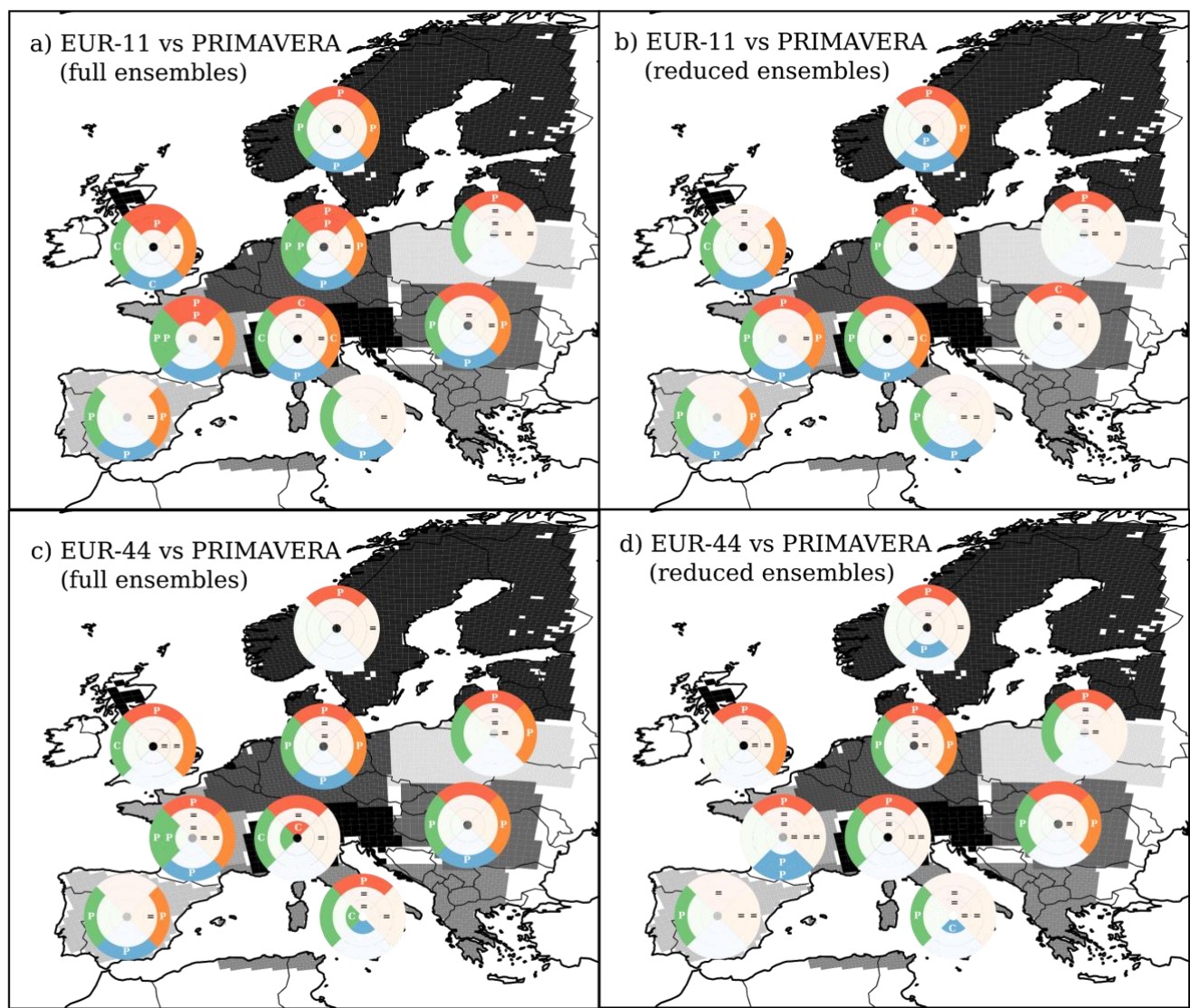

**Figure 8: Map using the method described in Fig. 1: for each region, season (clockwise from the top: summer, autumn, winter, spring; see right panel of Fig. 1), and precipitation intensity interval: low precipitation rates (inner part), moderate precipitation rates (middle part), high precipitation rates (outer part), a colour indicates that CORDEX and PRIMAVERA are significantly different, a "P" or "C" letter indicates that PRIMAVERA or CORDEX is closer to observations, respectively, an '=' sign indicates that both ensembles are close to observations. a) Map for the full PRIMAVERA and EUR-11 ensembles (listed in Tables 1 and 2); b) Map with reduced PRIMAVERA and EUR-11 ensembles using GCMs of the same family (coloured in Tables 1 and 2); c) Map for the full PRIMAVERA and EUR-44 ensembles (listed in Tables 1 and 2); d) Map with reduced PRIMAVERA and EUR-44 ensembles using GCMs of the same family (coloured in Tables 1 and 2).**