# Peer review of "European daily precipitation according to EURO-CORDEX RCMs"

_Geoscientific Model Development, 2019_

## Author Comment (AC1) · 9 Mar 2020

Please note that we have created a new DOI for the code and data availability sections: 10.5281/zenodo.3699683. This includes the analysis code, the regridding routines, and a /Data directory, which lists PIDs and tracking IDs (or filenames) of all data used.

---

## Referee Comment (RC1) · Anonymous Referee #1 · 30 Mar 2020

In the manuscript "Can high-resolution GCMs reach the level of information provided by 12-50 km CORDEX RCMs in terms of daily precipitation distribution?" Demory et al. evaluate precipitation distributions from an ensemble of regional climate model simulation with a novel ensemble of global high-resolution simulations. The authors aim to answer two questions: 1) If high-resolution GCMs can reach the quality of state-of-the-art RCMs in simulating precipitation over Europe? 2) Can climate modeling uncertainties be reduced by combining RCM and high-resolution GCM ensembles? The first question is well addressed and it is shown that precipitation from the new

high-resolution GCMs has a similar quality than precipitation from RCMs over Europe. This is a very promising result since it indicates that height-resolution GCMs might be able to provide high-quality precipitation simulations globally (or maybe at least in mid-latitudes). The second question is not addressed explicitly in the conclusion of the paper. The authors did an impressive job in evaluating a large number of models but the presentation and description of results are often vague and leaves a lot of room for interpretation. The novel part of this study is the comparison of RCMs with high-resolution GCMs and the paper would benefit from refocusing on this piece. The comparison of precipitation from 12 km and 50 km CORDEX simulations is already well documented and the sensitivity evaluation of the methods is important but takes up way too much space. Both topics draw the reader's attention away from the novel and innovative pieces of the manuscript. Furthermore, only precipitation intensity is evaluated and the simulation of precipitation patterns is neglected. I added more details about these and other issues and suggestions for improvements in the general and specific comments below. I am sorry that this review turned out to be so long but I see a lot of potential in this paper and sincerely tried to provide constructive guidance to help improve it.

General comments

1) I do not understand why the authors hypothesize that a high-resolution GCM should not have the same quality as an RCM downscaling a coarse resolution GCM. RCMs are tools that allow us to focus our available computational resources on a region of interest to derive higher-resolution climate information. The limiting factor here is the available computational resources. RCMs are not ultimately better in simulating regional climate than GCMs (although I agree that RCMs can be more tuned and are partly developed to perform well in a specific region). If we would have infinite computational resources we would not need RCMs anymore since a global high-resolution GCM should be able to outperform an RCM due to the potential to better simulate the global circulation and to close the water and energy cycle.

[Figure]

This is exactly the argument that Denis et al. (2002) used in their paper describing the so-called Big Brother Experiment (see section 2.1 in their paper). Therefore, I recommend rethinking the argumentation in your paper - starting with the title. One massive advantage of high-resolution GCMs, which is not discussed in the paper, is that they can provide high-resolution climate information globally. Therefore, the main conclusion of your paper for me (although you did not discuss this) is that we now have global models that have the potential of providing climate information that is on par with CORDEX downscaled datasets, which is a breakthrough.

This also shows that CORDEX has to evolve (which it does mainly through the flagship pilot studies) and explore new territory in climate modeling.

2) Related to the first comment, your paper reads biased. Especially in your introduction (e.g., L56), it reads like RCMs are superior tools compared to GCMs. However, RCMs have a major weakness, which is their dependence on lateral boundary conditions (LBC). Transitioning from a coarse-resolution GCM to a high-resolution RCM is not trivial and can cause problems. That is why RCMs use sponge zones and most studies typically do not show results close to the RCM computational domain due to massive biases, especially along the outflow boundaries. Furthermore, selecting an appropriate domain size and location is not trivial since too small domains cannot spin up higher-resolution features (e.g., Brisson et al. 2015) while too large domains can create unrealistic circulation patterns (Prein et al. 2019). A more unbiased discussion about the pros and cons of GCM and RCM modeling would be beneficial.

3) I suggest removing your "synthetic datasets" from your analysis. Simply multiplying observed precipitation by 1.2 is way too oversimplified to account for observational undercatch. If it would be that simple, the experts that create these observational datasets would have already done it. Precipitation undercatch is strongly dependent on the precipitation phase, intensity, the location of the stations (i.e., their exposure), and the type of gauge used (e.g., shielded vs. unshielded). It is fair to argue that observed precipitation is likely undercatched but adding 20 % more rainfall over the

entire distribution, in all regions, and all seasons is not scientifically justifiable.

Besides precipitation undercatch, there is another error source that affects specifically heavy rainfall in gridded observations, which is the soothing of intense rainfall over larger regions. This occurs in all gridded precipitation datasets that are purely based on gauges and results in dampening extreme precipitation peaks by smoothing the heavy rainfall and redistributing it larger regions. You can find more details about this in e.g., Haylock et al. (2008), Hofstra et al. (2009), or Prein and Gobiet (2017).

4) You are frequently using "resolution" when you refer to "horizontal grid spacing". E.g., the horizontal grid spacing of the CORDEX-44 simulations is roughly 50 km while their resolution is on the order of 2-8 time coarser (e.g., Skamarock 2004). The same concept applies to gridded observational datasets. Isotta et al. 2014 discuss that their precipitation dataset over the Alps has a grid spacing of 5 km while 10-15 km is a lower bound for the effective resolution of the dataset.

5) I appreciate that you performed a lot of sensitivity analysis to test the robustness of your results to various parameters. However, it takes you 1 ǍňΩ pages and two figures to show that your results are robust. Simply stating this in your method section and adding the figures to the supplement would be sufficient and would remove the distraction from your main results since you are not writing a methods paper but rather address impact researchers that want to use these climate simulations.

6) It is okay to assess the added value of simulated precipitation in the EURO-CORDEX-11 compare to the EURO-CORDEX-44 in this paper but I suggest to shorten this comparison since multiple papers have investigated this before and come to the same conclusions as you do (Kotlarski et al. 2014, Prein et al. 2016, Casanueva et al. 2016). This would allow the reader to focus more on the novel part of your analysis, which is the comparison of EURO-CORDEX with PRIMAVERA results.

7) You only evaluate precipitation distribution characteristics and neglect the spatial patterns of precipitation. Cutting back on the comparison between EURO-CORDEX

12 km and 50 km simulations and moving the sensitivity chapter to the supplement would free up enough space to add additional analysis that assesses the simulation of mean, and extreme (or light, moderate, and heavy) precipitation patterns. You could look at spatial correlation coefficients and standard deviations and create summary plots in the form of e.g., Taylor diagrams (Taylor 2001).

8) You do not directly address the two questions that you pose in L151-155 in your conclusions. Especially a discussion on your 2nd questions is missing.

9) The paper could be overall shortened, would benefit from restructuring, and can be in many places more precise. Also, a careful review of the language and style would improve the paper. I added multiple suggestions to the specific comments, which aim to address these issues, but I certainly did not capture everything.

Specific Comments:

L1: I suggest to use EURO-CORDEX instead of CORDEX in your title to make it clear that your analysis focuses on Europe.

L25: At this point in your paper it is not clear what you mean with rainfall bins. It would be better to talk about the contribution of various rainfall rates to total rainfall or something similar here.

L26: high-quality

L28: CMIP5 GCMs cannot capture because

L30: moderate-rainfall

L32-33: This is overly complicated. How about: Extreme precipitation simulated in PRIMAVERA GCMS agrees better with observations while CORDEX overestimates precipitation extremes.

L35-36: Please be careful with such broad statements since you do not assess any benefits for impact studies in your paper but you assess the simulation of precipitation.
L38: representation of the land-sea

L39-40: I think this is a misconception. I agree that it would be beneficial to better coordinate GCM and RCM efforts but RCMs will always be able to run at higher grid spacings than GCMs when using the same computational resources. RCMs should continue to explore the benefit of grid spacings that are out of reach for GCMs (e.g., kilometer-scale grid spacings), which can, in turn, inform GCM development.

L44: It would be good to add some references for RCM modeling here (e.g., Giorgi and Mearns 1991, Giorgi 2019).

L46: RCMs are also designed to balance the same resources.

L49: RCMs are not cheaper to run than atmosphere-only GCMs but they allow to focus resources on a region rather than having to simulate the entire globe.

L56: sample many different

L56-58: Your point 4 is more a disadvantage than an advantage. The need for RCMs introduces another layer of uncertainty, which is hard to quantify unless you downscale a large number of GCMs by a range of RCMs. Running high-resolution GCMs such as in the PRIMAVERA project should be an advantage especially since you show that the simulated precipitation quality is similar to the one from CORDEX without the extra need of downscaling.

L61: There are also regional earth system models, that can be as complex as ESMs (Zhang et al 2020).

L72: would be complementary

L76: focused its effort on downscaling

L79: I suggest to remove CORDEX CORE since it is not relevant for your analysis. You could mention it in the discussion.

L94: high extreme fall / also, what are Mediterranean events? Do you mean extreme precipitation events here?

L97: A discussion of one of the main results of Kotlarski et al. (2014) is missing which is "For seasonal mean quantities averaged over large European subdomains, no clear benefit of an increased spatial resolution (12 vs. 50 km) can be identified."

L101: RCMs are strongly constrained by

L105: should uncertainty be quality here?

L108: GCMs contribute to a lesser extend

L110-116: I do not understand why you introduce convection-resolving RCM simulations here. The discussion section would be a better place for this. Also, what do you mean by recently developed 2-step nesting convection-resolving simulations? 2-step nesting is not specific and not necessary for convection-resolving simulations and can also be applied in coarser resolution RCM simulations.

201: You mention IPCC already earlier.

L201: What is the grid spacing that you mention here? Some CMIP5 models already had 100 km horizontal grid spacing.

L225: atmosphere only

L130: You could mention the 25 km CAM experiments by Bacmeister et al. (2014) and Wehner et al (2014) here.

L131: What do you mean by "weather-type systems that feedback" here? I assume you mean the upscaling of local-scale features on the large-scale simulation.

L141: Similar to the above, what do you mean by "weather-type processes". The term weather type has a very specific meaning and refers to a specific weather pattern (see e.g., Philipp et al 2010).

L153-154: what do you mean with "spread of information"?

L168: which method are you talking about?

L176: "one per model" which one did you choose?

L179: should the statement "please refer to" be removed?

L198: "the best" it is unclear what you mean by this. This could mean various things. Did you mean "observations with the highest station density"?

L199: The ALPS-EURO4M dataset is not a national dataset and also CARPACLIM includes several countries.

L247: What is this EUR-44 rotated pole grid?

L250: It would be good to tell the reader what you are bootstrapping here.

L254: for the difference

L315: I suggest to move the discussion of Fig 8 and Fig. 8 here. Also, it would be interesting to remap the observations to a 12 km, 50 km, 100 km, and 200 km grid to understand if these differences are purely based on the coarser grid spacing or if the high-resolution models can add additional value.

L319: I would not say that deep convection schemes are more appropriate in RCMs since many of these schemes were developed for tropical convection in GCMs.

L321: reduced compared to DJF suggesting that such a resolution'

L331-332: similar compared to

L346-349: This text could be removed since it should be covered in the figure caption.

L354: "when the strict criteria" please be more specific here. E.g., remains true at the 99 % confidence level.

L361-362: central or eastern Europe

L366: ensembles differ most from the' Also, winter is the season with the largest precipitation undercatch in snow-dominated climates.

L440-441: This is a classic example where it is important to differentiate between model resolution and grid spacing. A model with semi-implicit semi-Lagrangian numerics might have the same grid spacing than a model with an e.g. split-explicit 3rd order Runge-Kutta time integration scheme but the latter will have a higher effective model resolution.

L459-460: I suggest to remove this sentence since it is very speculative.

L465-471: Relate to my general comment 5; This paragraph can be easily shortened to something like: "Our results are not sensitive to '"

L551-552: Why are you highlighting this? Many studies investigate the impact of aerosols on climate change.

L505: A clean comparison would only be possible if you would downscale a PRIMAVERA GCM with an RCM at the same grid spacing.

Literature:

Bacmeister, J.T., Wehner, M.F., Neale, R.B., Gettelman, A., Hannay, C., Lauritzen, P.H., Caron, J.M. and Truesdale, J.E., 2014. Exploratory high-resolution climate simulations using the Community Atmosphere Model (CAM). Journal of Climate, 27(9), pp.3073-3099.

Brisson, E., Demuzere, M. and Van Lipzig, N., 2015. Modelling strategies for performing convection-permitting climate simulations. Meteorologische Zeitschrift, 25(2), pp.149-163.

Casanueva, A., Kotlarski, S., Herrera, S., Fern$\sqrt{}$°ndez, J., Guti$\sqrt{}$©rrez, J.M., Boberg, F., Colette, A., Christensen, O.B., Goergen, K., Jacob, D. and Keuler, K., 2016. Daily precipitation statistics in a EURO-CORDEX RCM ensemble: added value of raw and

bias-corrected high-resolution simulations. Climate dynamics, 47(3-4), pp.719-737.

Denis, B., Laprise, R., Caya, D. and C√Âĕt√©, J., 2002. Downscaling ability of one-way nested regional climate models: the Big-Brother Experiment. Climate Dynamics, 18(8), pp.627-646. Giorgi, F. and Mearns, L.O., 1991. Approaches to the simulation of regional climate change: a review. Reviews of Geophysics, 29(2), pp.191-216.

Giorgi, F., 2019. Thirty years of regional climate modeling: where are we and where are we going next?. Journal of Geophysical Research: Atmospheres, 124(11), pp.5696-5723.

Haylock, M.R., Hofstra, N., Klein Tank, A.M.G., Klok, E.J., Jones, P.D. and New, M., 2008. A European daily high'Ăêresolution gridded data set of surface temperature and precipitation for 1950-2006. Journal of Geophysical Research: Atmospheres, 113(D20).

Hofstra, N., Haylock, M., New, M. and Jones, P.D., 2009. Testing E'ĂêOBS European high'Ăêresolution gridded data set of daily precipitation and surface temperature. Journal of Geophysical Research: Atmospheres, 114(D21).

Isotta, F.A., Frei, C., Weilguni, V., PerÆŠçec TadiÆŠá, M., Lassegues, P., Rudolf, B., Pavan, V., Cacciamani, C., Antolini, G., Ratto, S.M. and Munari, M., 2014. The climate of daily precipitation in the Alps: development and analysis of a high'Ăêresolution grid dataset from pan'ĂêAlpine rain'Ăêgauge data. International Journal of Climatology, 34(5), pp.1657-1675.

Kotlarski, S., Keuler, K., Christensen, O.B., Colette, A., Déqué, M., Gobiet, A., Goergen, K., Jacob, D., Láthi, D., Van Meijgaard, E. and Nikulin, G., 2014. Regional climate modeling on European scales: a joint standard evaluation of the EURO-CORDEX RCM ensemble. Geoscientific Model Development, 7, pp.1297-1333.

Philipp, A., Bartholy, J., Beck, C., Erpicum, M., Esteban, P., Fettweis, X., Huth, R., James, P., Jourdain, S., Kreienkamp, F. and Krennert, T., 2010. Cost733cat-A

database of weather and circulation type classifications. Physics and Chemistry of the Earth, Parts A/B/C, 35(9-12), pp.360-373.

Prein, A.F. and Gobiet, A., 2017. Impacts of uncertainties in European gridded precipitation observations on regional climate analysis. International Journal of Climatology, 37(1), pp.305-327.

Prein, A.F., Bukovsky, M.S., Mearns, L.O., Bruy$\sqrt{}$®re, C. and Done, J.M., 2019. Simulating North American Weather Types With Regional Climate Models. Frontiers in Environmental Science, 7, p.36.

Prein, A.F., Gobiet, A., Truhetz, H., Keuler, K., Goergen, K., Teichmann, C., Maule, C.F., Van Meijgaard, E., D$\sqrt{}$©qu$\sqrt{}$©, M., Nikulin, G. and Vautard, R., 2016. Precipitation in the EURO-CORDEX $ $0.11^{\circ} $$ and $ $0.44^{\circ} $ $0.44'àò simulations: high resolution, high benefits?. Climate dynamics, 46(1-2), pp.383-412.

Skamarock, W.C., 2004. Evaluating mesoscale NWP models using kinetic energy spectra. Monthly weather review, 132(12), pp.3019-3032.

Taylor, K.E., 2001. Summarizing multiple aspects of model performance in a single diagram. Journal of Geophysical Research: Atmospheres, 106(D7), pp.7183-7192.

Wehner, M.F., Reed, K.A., Li, F., Bacmeister, J., Chen, C.T., Paciorek, C., Gleckler, P.J., Sperber, K.R., Collins, W.D., Gettelman, A. and Jablonowski, C., 2014. The effect of horizontal resolution on simulation quality in the Community Atmospheric Model, CAM 5.1. Journal of Advances in Modeling Earth Systems, 6(4), pp.980-997.

Zhang, W., D$\sqrt{\partial}$scher, R., Koenigk, T., Miller, P.A., Jansson, C., Samuelsson, P., Wu, M. and Smith, B., 2020. The Interplay of Recent Vegetation and Sea Ice Dynamics-Results From a Regional Earth System Model Over the Arctic. Geophysical Research Letters, 47(6), p.e2019GL085982.

---

## Referee Comment (RC2) · Anonymous Referee #2 · 17 Apr 2020

Overarching Assessment

This paper compares high-resolution GCM simulations that were produced for High-ResMIP with Euro-CORDEX simulations at a similar resolution. The study is limited to a comparison of the daily precipitation distribution in the two ensembles. Overall, it is novel and interesting, but has a tendency towards making oversimplified statements that needs to be corrected.

General Comments

1. Without using additional observations, and without the presence of additional high-quality datasets for use, I understand that it is hard to include observational uncertainty. Applying a broad correction to illustrate this uncertainty is a little crude though. I do think it should be included, but lacking a better option, I think you need to better emphasize that this is by no means the ideal way to include an estimate of the observational uncertainty and it is also uncertain. Also, I think calling it a "synthetic dataset" is an unfortunate choice in terminology, as it is not fake or insincere, and I suggest you rethink that choice. At line 359, for example, you could simply say "They are further away from observations, but closer if our measure of uncertainty due to undercatch is considered."

2. You cite Roberts et al. 2018 (BAMS), but there's a point they make that I think needs to be highlighted in your discussion as well (see their paragraph that spans page 2342-2343). That is, that high-resolution GCMs are likely to provide improved information at the synoptic scale, and because an RCM's representation of the large-scale can only be as good as that from the GCM, this implies that high-resolution GCMs may provide better boundary conditions for even higher resolution RCMs. See also Gutowski et al. (https://doi.org/10.1175/BAMS-D-19-0113.1) for a similar, relevant follow on discussion to Roberts et al. This point would be relevant, for instance, around line 479, where you make a statement of expectations that I do not agree with and which needs references or a better backup discussion if you intend include it.

3. The authors pose 2 questions around line 150, but only ever address question 1. I suggest either addressing the 2nd question or stating here that this paper will only address question 1.

4. There are a number of statements made in the text that I think are too specific or overly general based on what I see in the presented data. See the "specific comments" section for instances that I particularly think need to be corrected.

Specific Comments

Title: I suggest you change the title to state that you are assessing European CORDEX

RCMs, as the conclusions could be very different over different regions.

Abstract: The abstract should state that the PRIMAVERA simulations are a part of HighResMip.

Line 25: For the abstract, I suggest removing the reference to bins and making the sentence "We perform this exercise for the distribution of daily precipitation over Europe..."

Line 34-35: Instead of "PRIMAVERA appear to be closer to observations. However, when we apply an averaged precipitation undercatch error of 20%, CORDEX become closer to these synthetic datasets.", you might just say that PRIMAVERA may be closer to the observations in this regard, but that uncertainty exists in the observations due to a potential undercatch error, especially in heavy precipitation. I suggest this, because the reader at this point does not understand why you have chosen the 20% value, and because I disagree with the terminology choice for "synthetic datasets".

Line 54: The word tuned is used differently by different people and parts of the community; therefore, I suggest you be more specific. I, for one, associate "tuning" with the modification of specific parameter numbers (e.g., changing grassland albedo for your region to something that is known to be more appropriate for your region). It might be more all-encompassing to say that parameterization schemes can be chosen based on their appropriateness for the region and tuning can be completed to better match regional observations. Or, more generally, that an RCM's configuration can be customized to focus on and best simulate the most relevant climate processes of a given region.

Line 59: Remove "so-called". It has two meanings, the latter of which is not appropriate here and could be misconstrued. 1: a common name for something. 2: a word that is used to describe something that is not suitable or not correct

Line 110: While I partly agree with this statement, it isn't universally correct. I would say that it has "not always facilitated the communication".

**[GMDD](https://gmd.copernicus.org/)**

Interactive
comment

Line 111: 2-step nesting isn't relevant to this sentence (and not always needed), and convection-resolving simulations are not new, although they have only more recently become more mainstream in climate. I suggest revising this sentence.

Line 118: this implies that RCMs have not also become more complex; however, they too have been moving towards increasing complexity (e.g. Turuncogle and Sannino 2017, https://doi.org/10.1007/s00382-016-3241-1)

Line 167: I do not understand this statement, please rephrase: "and concludes with an opening"

Fig. 1: In the caption you reference parts a and b, but there are no a and b labels in the figure. Please modify one or the other for consistency.

Line 250: it is stated that 1000 bootstrap samples are used. Did the authors test this number and its effect on the distributions generated? It is not a very large number of samples, especially considering the input data, and the statement at line 283 suggests to me that it is in fact much too low a number of samples to reasonably sample all combinations. Please justify the use of this particular number of samples or use many more.

Line 255: How were these bins chosen? Are they representative of percentiles or thresholds that impacts users are concerned about? They seem random to me, and 60mm/day seems quite high for the high category for some regions.

Line 270: you should state here if they are in the supplement or not shown.

Fig 4, and other similar figures. The legend placement is sloppy in some of these images. It should be placed where it doesn't cover data.

Line 330: I disagree with the statement that "there is no systematic difference". I see that there is a distinct systematic difference in most regions regardless of season, and so do you, clearly, because you discuss it in the next paragraph. P clearly has more light rain, C has more heavy rain, and the area under the curve is greater in C than P.

[Figure]

Please be more accurate.

Line 336: the introduction to this sentence is confusing. Do you mean to state that "PRIMAVERA still overestimates low intensity precipitation in all seasons and regions, like CMIP5, although to a lesser extent."? And if so, where's the figure that shows that? I can't tell that it is "to a lesser extent" from the current content.

Line 340: At this point in the text, the statement here is not proven. You should add that this will be discussed later.

Line 345: What is the sensitivity to the results in this paragraph to undercatch error?

Fig. 7. The C and P indicators on this figure are unfortunately too small for the resolution/quality of the PDF, and they are very hard to read, even when zoomed in, because they are so fuzzy. This needs to be fixed.

Line 381: A pie chart including the observational uncertainty would be an interesting addition to the supplementary material, and address my question at line 345.

Line 415: Given how similar PRIMAVERA and CORDEX-44 are, this statement is difficult to verify by just eye-balling the difference between this figure and previous figures. CORDEX-44 and CORDEX-11 are not identical and some of the small differences may matter. Could you overlay all 3 datasets?

Line 445: Not all convective precipitation schemes (CPS) take into account convective inhibition in their triggering function. This makes a difference in the drizzle problem. In my experience, RCMs are more likely to use CPS that include this (in mid-latitudes at least), and GCMs are less likely to. This isn't tuning, it's a configuration choice.

Line 483: "over CMIP5-driven CORDEX simulations for precipitation over Europe"... this is a bit general. It would be more accurate to say "over CMIP5-driven CORDEX simulations in some regions and seasons by our metrics for precipitation over Europe"

Line 486: Please re-write this sentence, I do not understand at all what it is trying to

say. And, if I guess at what it is trying to say, I do not agree with the statement.

Technical Corrections:

I do not think it should be the responsibility of the reviewer to copy-edit the manuscript. This paper is readable and well-organized, but could use some English language copy-editing. I have pointed out important instances of text in the previous section that I think need to be corrected for clarity purposes though, as these effect the understanding of the science presented. Please pay particular attention those.

---

## Author Comment (AC2) · 22 Jul 2020

Response to Anonymous Referee #1

We would like to thank the referee for the thorough review and constructive comments. We address the comments below. The reviewer's comments are in black, our responses in red.

In the manuscript "Can high-resolution GCMs reach the level of information provided by 12-50 km CORDEX RCMs in terms of daily precipitation distribution?" Demory et al. evaluate precipitation distributions from an ensemble of regional climate model simulation with a novel ensemble of global high-resolution simulations. The authors aim to answer two questions: 1) If high-resolution GCMs can reach the quality of state- of-the-art RCMs in simulating precipitation over Europe? 2) Can climate modeling uncertainties be reduced by combining RCM and high-resolution GCM ensembles? The first question is well addressed and it is shown that precipitation from the new high-resolution GCMs has a similar quality than precipitation from RCMs over Europe. This is a very promising result since it indicates that height-resolution GCMs might be able to provide high-quality precipitation simulations globally (or maybe at least in mid-latitudes). The second question is not addressed explicitly in the conclusion of the paper. The authors did an impressive job in evaluating a large number of models but the presentation and description of results are often vague and leaves a lot of room for interpretation. The novel part of this study is the comparison of RCMs with high-resolution GCMs and the paper would benefit from refocusing on this piece. The comparison of precipitation from 12 km and 50 km CORDEX simulations is already well documented and the sensitivity evaluation of the methods is important but takes up way too much space. Both topics draw the reader's attention away from the novel and innovative pieces of the manuscript. Furthermore, only precipitation intensity is evaluated and the simulation of precipitation patterns is neglected. I added more details about these and other issues and suggestions for improvements in the general and specific comments below. I am sorry that this review turned out to be so long but I see a lot of potential in this paper and sincerely tried to provide constructive guidance to help improve it.

We would like to thank you for your support in this study. We have revised the entire manuscript by focusing on the evaluation of high-resolution GCMs and CORDEX RCMs only, as suggested, and strengthened the description of the results and discussion. We have performed more analyses by showing spatial distribution of precipitation seasonal means for the two ensembles, as well as Taylor diagrams, which strengthen our conclusions. We agree that the comparison between EUR-11 and EUR-44 is well documented, so we have reduced that section. However, we now systematically include both EUR-11 and EUR-44 to the comparison with PRIMAVERA, with now an emphasis on EUR-11. While EUR-44 is closer to PRIMAVERA in terms of horizontal grid spacing, EUR-11 is the latest state-of-the-art EURO-CORDEX ensemble, which is more appropriate to compare with the state-of-the-art high-resolution GCM simulations (EUR-44 is a bit older). We have also increased the size of the EUR-11 ensemble, as more simulations have now become available, giving an overview of the performance of the latest EURO-CORDEX ensemble. We have removed what we called the "synthetic datasets", as suggested, but have included a discussion section on observational uncertainty. Finally, we have moved the figures related to the sensitivity tests to additional material and have reduced the size of these sections, as suggested. To represent the ensemble spread, we now use the inter-quartile range (not a bootstrap resampling anymore), as it is more representative of the ensemble size. And we now use a student t-test to build the pies, instead of p-values. More details are given below in response to each comment.

General comments

1) I do not understand why the authors hypothesize that a high-resolution GCM should not have the same quality as an RCM downscaling a coarse resolution GCM. RCMs are tools that allow us to focus our available computational resources on a region of interest to derive higher-resolution climate information. The limiting factor here is the available computational resources. RCMs are not ultimately better in simulating regional climate than GCMs (although I agree that RCMs can be more tuned and are partly developed to perform well in a specific region). If we would have infinite computational resources we would not need RCMs anymore since a global high-resolution GCM should be able to outperform an RCM due to the potential to better simulate the global circulation and to close the water and energy cycle.

This is exactly the argument that Denis et al. (2002) used in their paper describing the so-called Big Brother Experiment (see section 2.1 in their paper). Therefore, I recommend rethinking the argumentation in your paper - starting with the title.

Thank you for this comment. We have now revised our title, introduction and discussion entirely. We agree that we should not make hypotheses whether GCM or RCM would be better at providing regional climate information, and rather list pros and cons of each approach, which we now do in the introduction.

One massive advantage of high-resolution GCMs, which is not discussed in the paper, is that they can provide high-resolution climate information globally. Therefore, the main conclusion of your paper for me (although you did not discuss this) is that we now have global models that have the potential of providing climate information that is on par with CORDEX downscaled datasets, which is a breakthrough.

This also shows that CORDEX has to evolve (which it does mainly through the flagship pilot studies) and explore new territory in climate modeling.

We would like to thank the reviewer for this suggestion. We added this point to our discussion and the evolution of CORDEX towards km-scale resolution is also discussed in a final section on *Opportunities for future coordination*.

2) Related to the first comment, your paper reads biased. Especially in your introduction (e.g., L56), it reads like RCMs are superior tools compared to GCMs. However, RCMs have a major weakness, which is their dependence on lateral boundary conditions (LBC). Transitioning from a coarse-resolution GCM to a high-resolution RCM is not trivial and can cause problems. That is why RCMs use sponge zones and most studies typically do not show results close to the RCM computational domain due to massive biases, especially along the outflow boundaries. Furthermore, selecting an appropriate domain size and location is not trivial since too small domains cannot spin up higher-resolution features (e.g., Brisson et al. 2015) while too large domains can create unrealistic circulation patterns (Prein et al. 2019). A more unbiased discussion about the pros and cons of GCM and RCM modeling would be beneficial.

We would like to thank the reviewer for this suggestion and references. We have altered the introduction accordingly and have added the suggested references, together with Denis et al, 2002, listed in the previous comment.

3) I suggest removing your "synthetic datasets" from your analysis. Simply multiplying observed precipitation by 1.2 is way too oversimplified to account for observational undercatch. If it would be that simple, the experts that create these observational datasets would have already done it. Precipitation undercatch is strongly dependent on the precipitation phase, intensity, the location of the stations (i.e., their exposure), and the type of gauge used (e.g., shielded vs. unshielded). It is fair to argue that observed precipitation is likely undercatched but adding 20 % more rainfall over the entire distribution, in all regions, and all seasons is not scientifically justifiable.

We have now added a new section "Observational uncertainty" (section 4.3) that describes pros and cons of these gridded observational datasets, as suggested above. We also followed the advice to remove the "synthetic datasets", and only kept a figure in the supplementary material (Fig. S7), which was suggested by reviewer #2.

Besides precipitation undercatch, there is another error source that affects specifically heavy rainfall in gridded observations, which is the soothing of intense rainfall over larger regions. This occurs in all gridded precipitation datasets that are purely based on gauges and results in dampening extreme precipitation peaks by smoothing the heavy rainfall and redistributing it larger regions. You can find more details about this in e.g., Haylock et al. (2008), Hofstra et al. (2009), or Prein and Gobiet (2017).

Thank you for this suggestion. We have added this additional information and references in the new subsection 4.3.

4) You are frequently using "resolution" when you refer to "horizontal grid spacing". E.g., the horizontal grid spacing of the CORDEX-44 simulations is roughly 50 km while their resolution is on the order of 2-8 time coarser (e.g., Skamarock 2004). The same concept applies to gridded observational datasets. Isotta et al. 2014 discuss that their precipitation dataset over the Alps has a grid spacing of 5 km while 10-15 km is a lower bound for the effective resolution of the dataset.

We understand that the term "resolution" could be understood as the effective resolution, which is not the case here. As suggested, we have replaced the word "resolution" by "horizontal grid spacing" whenever possible throughout the manuscript, or used the term "horizontal resolution" to avoid confusion with effective resolution.

5) I appreciate that you performed a lot of sensitivity analysis to test the robustness of your results to various parameters. However, it takes you 1 Ân˘Ω pages and two figures to show that your results are robust. Simply stating this in your method section and adding the figures to the supplement would be sufficient and would remove the distraction from your main results since you are not writing a methods paper but rather address impact researchers that want to use these climate simulations.

We changed the structure of the article and moved the sensitivity tests figures to the supplementary material. We changed section 4 into "Evaluation of the robustness of results"

and now only discuss elements relevant to the results (Use of different thresholds to define significance, use of EUR-11 vs EUR-44 and observational uncertainty).

6) It is okay to assess the added value of simulated precipitation in the EURO- CORDEX-11 compare to the EURO-CORDEX-44 in this paper but I suggest to shorten this comparison since multiple papers have investigated this before and come to the same conclusions as you do (Kotlarski et al. 2014, Prein et al. 2016, Casanueva et al. 2016). This would allow the reader to focus more on the novel part of your analysis, which is the comparison of EURO-CORDEX with PRIMAVERA results.

Now that we have a larger ensemble with EUR-11 (as more EUR-11 simulations have become available on ESGF), we focus on EUR-11 to compare state-of-the-art GCM and RCM simulations, while EUR-44 is slightly older. However, as EUR-44 uses a grid spacing which is closer to PRIMAVERA, we also systematically include EUR-44 in our analyses whenever possible without making the figures too crowded. We can therefore have a systematic analysis of EUR-11, EUR-44 and PRIMAVERA. This is detailed in section 4.2, where we also discuss our results with respect to previous findings.

7) You only evaluate precipitation distribution characteristics and neglect the spatial patterns of precipitation. Cutting back on the comparison between EURO-CORDEX 12 km and 50 km simulations and moving the sensitivity chapter to the supplement would free up enough space to add additional analysis that assesses the simulation of mean, and extreme (or light, moderate, and heavy) precipitation patterns. You could look at spatial correlation coefficients and standard deviations and create summary plots in the form of e.g., Taylor diagrams (Taylor 2001).

We have now included a new section (3.2 Mean differences between EURO-CORDEX and PRIMAVERA ensembles) to discuss spatial distribution of mean biases and Taylor diagrams on seasonal mean precipitation (Fig 4 and 5 in the revised manuscript).

All these new results confirm the main findings of our study: PRIMAVERA generally performs better or similar to EURO-CORDEX. It also adds new information that, although EUR-44 generally has a small precipitation seasonal mean bias, its spatial distribution is further away from observations compared to PRIMAVERA, although they use similar horizontal grid spacings. We have added these new results in section 3.2 and discuss them with respect to previous studies comparing EUR-44 and EUR-11 in section 4.2.

8) You do not directly address the two questions that you pose in L151-155 in your conclusions. Especially a discussion on your 2nd questions is missing.

We agree with this comment and have removed question 2.

9) The paper could be overall shortened, would benefit from restructuring, and can be in many places more precise. Also, a careful review of the language and style would improve the paper. I added multiple suggestions to the specific comments, which aim to address these issues, but I certainly did not capture everything.

We have revised the manuscript entirely and have restructured it. We also show 8 figures now instead of 10, and moved the figures that are not directly linked to the main question of the study to additional material.

Specific Comments:

L1: I suggest to use EURO-CORDEX instead of CORDEX in your title to make it clear that your analysis focuses on Europe.

Done.

L25: At this point in your paper it is not clear what you mean with rainfall bins. It would be better to talk about the contribution of various rainfall rates to total rainfall or something similar here.

We have modified that sentence.

L26: high-quality

Done.

L28: CMIP5 GCMs cannot capture because

Done.

L30: moderate-rainfall

We have changed it to 'moderate-precipitation'.

L32-33: This is overly complicated. How about: Extreme precipitation simulated in PRIMAVERA GCMS agrees better with observations while CORDEX overestimates precipitation extremes.

Rephrased as suggested.

L35-36: Please be careful with such broad statements since you do not assess any benefits for impact studies in your paper but you assess the simulation of precipitation.

This sentence has been removed.

L38: representation of the land-sea

This has been removed.

L39-40: I think this is a misconception. I agree that it would be beneficial to better coordinate GCM and RCM efforts but RCMs will always be able to run at higher grid spacings than GCMs when using the same computational resources. RCMs should continue to explore the benefit of grid spacings that are out of reach for GCMs (e.g., kilometer-scale grid spacings), which can, in turn, inform GCM development.

We agree. By construction, RCMs will always be able to run at smaller grid spacing than GCMs for a given computational power. However, one must distinguish ordinary "production" runs from "cutting-edge" research runs. In this sense, cutting-edge GCM runs such as those from HiResMIP/PRIMAVERA have met ordinary production CORDEX RCM runs. As mentioned by the reviewer, CORDEX cutting-edge is at the kilometer-scale, dealing with resolved deep convection beyond the "grey-zone". And standard GCM runs are still carried out at grid spacings of ~100km (ScenarioMIP, DCPP) where they can afford to run many members to explore scenarios, internal variability and other uncertainties. Likewise, standard RCM runs are still in the range 10-50 km, where simulations for different driving GCMs and scenarios are affordable. Given that regional climate modelling already started with a grid spacing of ~60 km (Giorgi & Bates, 1989), this is the first time in history that an ensemble of GCMs reaches the resolution of RCM ensembles. Results should be compared, as we do in this work, but this is just the first step. There is a lot of room for improving GCM assessments, useful for both communities. This should be done with better coordination among the communities, as the cutting-edge and production runs of both communities will meet continuously in the future.

This was the motivation behind this last sentence in the abstract. We have removed this sentence from the abstract and added this discussion to the final section 5.2 with a discussion on "opportunities for future coordination".

L44: It would be good to add some references for RCM modeling here (e.g., Giorgi and Mearns 1991, Giorgi 2019).

Done.

L46: RCMs are also designed to balance the same resources. / L49: RCMs are not cheaper to run than atmosphere-only GCMs but they allow to focus resources on a region rather than having to simulate the entire globe. / L56: sample many different / L56-58: Your point 4 is more a disadvantage than an advantage. The need for RCMs introduces another layer of uncertainty, which is hard to quantify unless you downscale a large number of GCMs by a range of RCMs. Running high-resolution GCMs such as in the PRIMAVERA project should be an advantage especially since you show that the simulated precipitation quality is similar to the one from CORDEX without the extra need of downscaling. / L61: There are also regional earth system models, that can be as complex as ESMs (Zhang et al 2020).

The whole paragraph has been fully rewritten to address these specific comments and the more general comments 1) and 2).

L72: would be complementary

Changed.

L76: focused its effort on downscaling

Changed.

L79: I suggest to remove CORDEX CORE since it is not relevant for your analysis. You could mention it in the discussion.

We followed this suggestion. It is now mentioned in section 5.2.

L94: high extreme fall / also, what are Mediterranean events? Do you mean extreme precipitation events here?

Yes, it is now rephrased.

L97: A discussion of one of the main results of Kotlarski et al. (2014) is missing which is "For seasonal mean quantities averaged over large European subdomains, no clear benefit of an increased spatial resolution (12 vs. 50 km) can be identified."

This sentence has been rephrased to state that there is no clear benefit, and a discussion on the benefit of increasing resolution in EURO-CORDEX simulations compared to previous studies has been added in section 4.2.

L101: RCMs are strongly constrained by

Done.

L105: should uncertainty be quality here?

It would also fit. We changed it as suggested.

L108: GCMs contribute to a lesser extend

We have changed this sentence to 'biases in simulated radiation and surface wind speeds appear to be more related to the RCM than the driving GCM'.

L110-116: I do not understand why you introduce convection-resolving RCM simula- tions here. The discussion section would be a better place for this. Also, what do you mean by recently developed 2-step nesting convection-resolving simulations? 2-step nesting is not specific and not necessary for convection-resolving simulations and can also be applied in coarser resolution RCM simulations.

Agreed. Convection-resolving simulations have been moved to the discussion section 5.2 (as the new grid spacing target of RCMs) and the 2-step nesting mention removed.

201: You mention IPCC already earlier.

This should be L121 (not 201). Solved. It is not mentioned earlier now that the CORDEX-CORE has been moved to the discussion section.

L201: What is the grid spacing that you mention here? Some CMIP5 models already had 100 km horizontal grid spacing.

This should be L121 (not 201). This is the average resolution. Some also had much lower resolution than 150 km in CMIP5. We have added the term 'typically'.

L225: atmosphere only

This should be L125 (not 225). Done.

L130: You could mention the 25 km CAM experiments by Bacmeister et al. (2014) and Wehner et al (2014) here.

Done.

L131: What do you mean by "weather-type systems that feedback" here? I assume you mean the upscaling of local-scale features on the large-scale simulation.

We have replaced this sentence by "have drawn similar conclusions regarding the interactions (or feedbacks) between the synoptic scales and the large-scale global climate system".

L141: Similar to the above, what do you mean by "weather-type processes". The term weather type has a very specific meaning and refers to a specific weather pattern (see e.g., Philipp et al 2010).

We have replaced it by "synoptic-scale processes".

L153-154: what do you mean with "spread of information"?

We have removed that paragraph.

L168: which method are you talking about?

This relates to L169: "Method and Data" section. Method refers to sub-sections 2.7 and 2.8. We have kept it.

L176: "one per model" which one did you choose?

We usually consider the first available (r1). The chosen members are listed in Table 1, and we have added a reference to that table in this sentence.

L179: should the statement "please refer to" be removed?

We have removed that reference because this list is not updated anymore.

L198: "the best" it is unclear what you mean by this. This could mean various things. Did you mean "observations with the highest station density"?

Yes, we have replaced that sentence by 'Over Europe, we make use of high spatial resolution gridded observational datasets that include the highest station density'.

L199: The ALPS-EURO4M dataset is not a national dataset and also CARPACLIM includes several countries.

We removed the term 'national dataset' throughout the manuscript.

L247: What is this EUR-44 rotated pole grid?

The CORDEX domains are defined in rotated-pole coordinates. The rotated pole grid is therefore the standard projection for CORDEX RCMs.

L250: It would be good to tell the reader what you are bootstrapping here.

We have now changed this paragraph. We do not use the bootstrap resampling method anymore, but the inter-quartile range, which is more appropriate to use with ensembles of different sizes. Please refer to the response to reviewer #2 for Line 250.

L254: for the difference

We have changed that paragraph.

L315: I suggest to move the discussion of Fig 8 and Fig. 8 here. Also, it would be interesting to remap the observations to a 12 km, 50 km, 100 km, and 200 km grid to understand if these differences are purely based on the coarser grid spacing or if the high-resolution models can add additional value.

This comment refers to Fig. 9 and 10 of the submitted manuscript. Regridding observations at lower resolution than 50 km is out of scope of this study because we focus our analyses on the simulation of precipitation distribution in EURO-CORDEX (both EUR-11 and EUR-44) and PRIMAVERA at 50km scale. As the comparison between EUR-11 and EUR-44 has also been studied extensively, we have decided to remove Fig. 9 and 10 from the main article and replaced them by pie plots of PRIMAVERA vs EUR-44 and PRIMAVERA vs EUR-11@50km (Fig. 8). In Fig. 2 and 3 now showing CMIP5, EUR-11 and PRIMAVERA, we have added observations remapped at a 50km scale as suggested by Reviewer 2 (comment L336). These figures show that CMIP5 is not able to reproduce a proper distribution of precipitation. Torma et al., 2015 (see reference in the paper) did some tests by regridding the Alpine observations on 0.11, 0.44 and 1.32 degrees to compare CMIP5, EUR-11 and EUR-44 over the Alps. It is clear from their results (their Fig 6) that regridding observations to lower CMIP-type resolution changes the distribution over complex orography. Observations remain on the more intense side though, as does EUR-11.

To evaluate the impact of regridding, we have done some tests with EUR-11 on their native grid and regridded at 50km using the high-resolution observations remapped on EUR-11 and EUR-44. The figures are below. Note that these are based on the previous ensemble of EUR-11 (used for the submission) so cannot be compared with the new figures. It shows here that for all regions, regridding the observations slightly changes the distribution but remains within the interannual variability. Similarly, the differences between EUR-11 regridded on EUR-44 and EUR-11 on its native grid show a slight shift of the distribution towards higher intensity over most regions. We suspect that the large differences in both EUR-11 and observations over BI are due to the mask. These results show that whether we consider EUR-11 and observations on a common EUR-11 grid or EUR-44 grid would not change the main conclusion of our study. A similar conclusion has also been drawn by Iles et al., 2019. This is now discussed in section 4.2.

[Figure]

L319: I would not say that deep convection schemes are more appropriate in RCMs since many of these schemes were developed for tropical convection in GCMs.

We have removed it.

L321: reduced compared to DJF suggesting that such a resolution

We have removed this sentence, as we now compare EUR-44 and EUR-11 in section 4.2.

L331-332: similar compared to

We have changed that paragraph.

L346-349: This text could be removed since it should be covered in the figure caption.

Done.

L354: "when the strict criteria" please be more specific here. E.g., remains true at the 99 % confidence level.

We've restructured the text and made sure we were more specific in section 4.1.

L361-362: central or eastern Europe

We have changed this paragraph.

L366: ensembles differ most from the âA Ž Also, winter is the season with the largest precipitation undercatch in snow-dominated climates.

We have changed this paragraph and included the suggestion regarding precipitation undercatch in winter.

L440-441: This is a classic example where it is important to differentiate between model resolution and grid spacing. A model with semi-implicit semi-Lagrangian numerics might have the same grid spacing than a model with an e.g. split-explicit 3rd order Runge-Kutta time integration scheme but the latter will have a higher effective model resolution.

We now mention the potentially different effective resolution in the last paragraph of section 2.2.

L459-460: I suggest to remove this sentence since it is very speculative.

Done.

L465-471: Relate to my general comment 5; This paragraph can be easily shortened to something like: "Our results are not sensitive to âA Ž"

We removed the paragraph.

L551-552: Why are you highlighting this? Many studies investigate the impact of aerosols on climate change.

This refers to L501-502 of the submitted manuscript (not L551-552). We agree many studies investigate the impact of aerosols on climate change. We highlighted these two (Boe et al, 2020 and Gutierrez et al, 2020) because they explicitly discuss the different treatments of aerosols in GCMs and RCMs and how they impact climate change projections in each ensemble. The text has been rephrased to make this clear.

L505: A clean comparison would only be possible if you would downscale a PRIMAV- ERA GCM with an RCM at the same grid spacing.

We assume this comment refers to L183-184 of the submitted manuscript "We focus our analysis on the EUR-44 simulations because their resolution roughly corresponds to the resolutions used by PRIMAVERA GCMs, which allows a clean comparison between the two ensembles." We agree with this comment. Moreover, we now do not focus on EUR-44 only but also systematically include EUR-11. We have therefore modified the sentence by "The advantage of considering both EUR-44 and EUR-11 is that EUR-44 RCMs' horizontal grid spacings roughly correspond to that of PRIMAVERA GCMs, and EUR-11 is based on state-of-the-art model generations (EUR-44 is slightly older). These two EURO-CORDEX ensembles make the comparison with state-of-the-art high-resolution GCM simulations more appropriate."

Literature:

Bacmeister, J.T., Wehner, M.F., Neale, R.B., Gettelman, A., Hannay, C., Lauritzen, P.H., Caron, J.M. and Truesdale, J.E., 2014. Exploratory high-resolution climate simulations using the Community Atmosphere Model (CAM). Journal of Climate, 27(9), pp.3073- 3099.

Brisson, E., Demuzere, M. and Van Lipzig, N., 2015. Modelling strategies for per- forming convection-permitting climate simulations. Meteorologische Zeitschrift, 25(2), pp.149-163.

Casanueva, A., Kotlarski, S., Herrera, S., Fern√∘ndez, J., Guti√©rrez, J.M., Boberg, F., Colette, A., Christensen, O.B., Goergen, K., Jacob, D. and Keuler, K., 2016. Daily precipitation statistics in a EURO-CORDEX RCM ensemble: added value of raw and bias-corrected high-resolution simulations. Climate dynamics, 47(3-4), pp.719-737.

Denis, B., Laprise, R., Caya, D. and C√Âeˇt√©, J., 2002. Downscaling ability of one- way nested regional climate models: the Big-Brother Experiment. Climate Dynamics, 18(8), pp.627-646. Giorgi, F. and Mearns, L.O., 1991. Approaches to the simulation of regional climate change: a review. Reviews of Geophysics, 29(2), pp.191-216.

Giorgi, F., 2019. Thirty years of regional climate modeling: where are we and where are we going next?. Journal of Geophysical Research: Atmospheres, 124(11), pp.5696- 5723.

Haylock, M.R., Hofstra, N., Klein Tank, A.M.G., Klok, E.J., Jones, P.D. and New, M., 2008. A European daily high'Äêresolution gridded data set of surface tempera- ture and precipitation for 1950-2006. Journal of Geophysical Research: Atmospheres, 113(D20).

Hofstra,N.,Haylock,M.,New,M.andJones,P.D.,2009.TestingE'ÄêOBSEuro- peanhigh'Äêresolutiongriddeddatasetofdailyprecipitationandsurfacetempera- ture. Journal of Geophysical Research: Atmospheres, 114(D21).

Isotta, F.A., Frei, C., Weilguni, V., PerÆŠçec TadiÆŠá, M., Lassegues, P., Rudolf, B., Pavan, V., Cacciamani, C., Antolini, G., Ratto, S.M. and Munari, M., 2014. The climate ofdailyprecipitationintheAlps:developmentandanalysisofahighâA ŽÄêresolution griddatasetfrompanâA ŽÄêAlpinerainâA ŽÄêgaugedata.InternationalJournalofCli- matology, 34(5), pp.1657-1675.

Kotlarski, S., Keuler, K., Christensen, O.B., Colette, A., D√©qu√©, M., Gobiet, A., Go- ergen, K., Jacob, D., L√◦thi, D., Van Meijgaard, E. and Nikulin, G., 2014. Regional cli- mate modeling on European scales: a joint standard evaluation of the EURO-CORDEX RCM ensemble. Geoscientific Model Development, 7, pp.1297-1333.

Philipp, A., Bartholy, J., Beck, C., Erpicum, M., Esteban, P., Fettweis, X., Huth, R., James, P., Jourdain, S., Kreienkamp, F. and Krennert, T., 2010. Cost733cat-A database of weather and circulation type classifications. Physics and Chemistry of the Earth, Parts A/B/C, 35(9-12), pp.360-373.

Prein, A.F. and Gobiet, A., 2017. Impacts of uncertainties in European gridded precip- itation observations on regional climate analysis. International Journal of Climatology, 37(1), pp.305-327.

Prein, A.F., Bukovsky, M.S., Mearns, L.O., Bruyère, C. and Done, J.M., 2019. Sim- ulating North American Weather Types With Regional Climate Models. Frontiers in Environmental Science, 7, p.36.

Prein, A.F., Gobiet, A., Truhetz, H., Keuler, K., Goergen, K., Teichmann, C., Maule, C.F., Van Meijgaard, E., D√©qu√©, M., Nikulin, G. and Vautard, R., 2016. Precipitation in theEURO-CORDEX$$0.11ˆ{\circ}$$and$$0.44ˆ{\circ}$$0.44âA Žàòsimulations: high resolution, high benefits?. Climate dynamics, 46(1-2), pp.383-412.

Skamarock, W.C., 2004. Evaluating mesoscale NWP models using kinetic energy spectra. Monthly weather review, 132(12), pp.3019-3032.

Taylor, K.E., 2001. Summarizing multiple aspects of model performance in a single diagram. Journal of Geophysical Research: Atmospheres, 106(D7), pp.7183-7192.

Wehner, M.F., Reed, K.A., Li, F., Bacmeister, J., Chen, C.T., Paciorek, C., Gleckler, P.J., Sperber, K.R., Collins, W.D., Gettelman, A. and Jablonowski, C., 2014. The effect of horizontal resolution on simulation quality in the Community Atmospheric Model, CAM 5.1. Journal of Advances in Modeling Earth Systems, 6(4), pp.980-997.

Zhang, W., Döscher, R., Koenigk, T., Miller, P.A., Jansson, C., Samuelsson, P., Wu, M. and Smith, B., 2020. The Interplay of Recent Vegetation and Sea Ice Dynamics- Results From a Regional Earth System Model Over the Arctic. Geophysical Research Letters, 47(6), p.e2019GL085982.

---

## Author Comment (AC3)

Response to Anonymous Referee #2

We would like to thank the referee for the thorough review and constructive comments. We address the comments below. The reviewer's comments are in black, our responses in red.

Overarching Assessment

This paper compares high-resolution GCM simulations that were produced for High- ResMIP with Euro-CORDEX simulations at a similar resolution. The study is limited to a comparison of the daily precipitation distribution in the two ensembles. Overall, it is novel and interesting, but has a tendency towards making oversimplified statements that needs to be corrected.

We have revised the entire manuscript. We have performed more analyses by showing spatial distribution of precipitation seasonal means for the two ensembles, as well as Taylor diagrams, as suggested by Reviewer 1, which strengthen our conclusions. We also now systematically include both EUR-11 and EUR-44 to the comparison with PRIMAVERA, with now an emphasis on EUR-11. While EUR-44 is closer to PRIMAVERA in terms of horizontal grid spacing, EUR-11 is the latest state-of-the-art EURO-CORDEX ensemble, which is more appropriate to compare with the state-of-the-art high-resolution GCM simulations (EUR-44 is a bit older). We have also increased the size of the EUR-11 ensemble, as more simulations have now become available, giving an overview of the performance of the latest EURO-CORDEX ensemble. We have removed what we called the "synthetic datasets", but have included a discussion section on observational uncertainty. Finally, we have moved the figures related to the sensitivity tests to additional material. To represent the ensemble spread, we now use the inter-quartile range (not a bootstrap resampling anymore), as it is more representative of the ensemble size. And we now use a student t-test to build the pies, instead of p-values. More details are given below in response to each comment.

General Comments

1. Without using additional observations, and without the presence of additional high- quality datasets for use, I understand that it is hard to include observational uncertainty. Applying a broad correction to illustrate this uncertainty is a little crude though. I do think it should be included, but lacking a better option, I think you need to better empha- size that this is by no means the ideal way to include an estimate of the observational uncertainty and it is also uncertain. Also, I think calling it a "synthetic dataset" is an un- fortunate choice in terminology, as it is not fake or insincere, and I suggest you rethink that choice. At line 359, for example, you could simply say "They are further away from observations, but closer if our measure of uncertainty due to undercatch is considered."

We would like to thank the reviewer for this suggestion. As also suggested by Reviewer 1, we have now removed these "synthetic datasets" from our analyses. We still keep them in a pie-chart figure in the supplementary material to discuss observational uncertainty (section 4.2).

2. You cite Roberts et al. 2018 (BAMS), but there's a point they make that I think needs to be highlighted in your discussion as well (see their paragraph that spans page 2342- 2343). That is, that high-resolution GCMs are likely to provide improved information at the synoptic scale, and because an RCM's representation of the large-scale can only be as good as that from the GCM, this implies that high-resolution GCMs may provide better boundary

conditions for even higher resolution RCMs. See also Gutowski et al. (https://doi.org/10.1175/BAMS-D-19-0113.1) for a similar, relevant follow on discussion to Roberts et al. This point would be relevant, for instance, around line 479, where you make a statement of expectations that I do not agree with and which needs references or a better backup discussion if you intend include it.

We have decided to keep the sentence 'The performance of PRIMAVERA was not logically expected', but we have reformulated the paragraph and now provide a discussion regarding this statement. We also now include a reference to Gutowski et al, thank you for pointing to it. It is now written in the first paragraph of section 5.2.

3. The authors pose 2 questions around line 150, but only ever address question 1. I suggest either addressing the 2nd question or stating here that this paper will only address question 1.

We have changed this paragraph and now only specifically address question 1.

4. There are a number of statements made in the text that I think are too specific or overly general based on what I see in the presented data. See the "specific comments" section for instances that I particularly think need to be corrected.

This issue was also reported by Reviewer 1. We have revised our entire results and discussion sections.

Specific Comments

Title: I suggest you change the title to state that you are assessing European CORDEX RCMs, as the conclusions could be very different over different regions.

Done.

Abstract: The abstract should state that the PRIMAVERA simulations are a part of HighResMip.

Done.

Line 25: For the abstract, I suggest removing the reference to bins and making the sentence "We perform this exercise for the distribution of daily precipitation over Europe. . ."

We have replaced that sentence by 'The focus of the evaluation is on the distribution of daily precipitation at 50km scale under current climate conditions'.

Line 34-35: Instead of "PRIMAVERA appear to be closer to observations. However, when we apply an averaged precipitation undercatch error of 20%, CORDEX become closer to these synthetic datasets.", you might just say that PRIMAVERA may be closer to the observations in this regard, but that uncertainty exists in the observations due to a potential undercatch error, especially in heavy precipitation. I suggest this, because the reader at this point does not understand why you have chosen the 20% value, and because I disagree with the terminology choice for "synthetic datasets".

Rephrased as suggested.

Line 54: The word tuned is used differently by different people and parts of the com- munity; therefore, I suggest you be more specific. I, for one, associate "tuning" with the modification of specific parameter numbers (e.g., changing grassland albedo for your region to something that is known to be more appropriate for your region). It might be more all-encompassing to say that parameterization schemes can be chosen based on their appropriateness for the region and tuning can be completed to better match regional observations. Or, more generally, that an RCM's configuration can be cus- tomized to focus on and best simulate the most relevant climate processes of a given region.

The sentence was rephrased as suggested.

Line 59: Remove "so-called". It has two meanings, the latter of which is not appropriate here and could be misconstrued. 1: a common name for something. 2: a word that is used to describe something that is not suitable or not correct

Agreed. We just referred to the common name for these models, so we removed it.

Line 110: While I partly agree with this statement, it isn't universally correct. I would say that it has "not always facilitated the communication".

We have removed that sentence.

Line 111: 2-step nesting isn't relevant to this sentence (and not always needed), and convection-resolving simulations are not new, although they have only more recently become more mainstream in climate. I suggest revising this sentence.

We agree and have now removed this paragraph. We now mention the convection-resolving simulations in the discussion section (5.2) as the new target for future RCM simulations.

Line 118: this implies that RCMs have not also become more complex; however, they too have been moving towards increasing complexity (e.g. Turuncogle and Sannino 2017, https://doi.org/10.1007/s00382-016-3241-1)

Agreed. The sentence does not mention RCMs anymore and we now also mention regional ESM earlier on together with the suggested reference, thank you.

Line 167: I do not understand this statement, please rephrase: "and concludes with an opening"

This sentence has been removed.

Fig. 1: In the caption you reference parts a and b, but there are no a and b labels in the figure. Please modify one or the other for consistency.

Thank you for spotting this. We have changed the caption accordingly.

Line 250: it is stated that 1000 bootstrap samples are used. Did the authors test this number and its effect on the distributions generated? It is not a very large number of samples,

especially considering the input data, and the statement at line 283 suggests to me that it is in fact much too low a number of samples to reasonably sample all combinations. Please justify the use of this particular number of samples or use many more.

We tried two numbers of samples: $10_3$ and $10_6$. Using $10_6$ did not particularly change the results, and it did not change our main conclusions. However, we understand the concern and now that we include more EUR-11 simulations, which include ~40 members, the number of combinations becomes very large, and we agree that the bootstrapping method may reach its limits. We have therefore replaced the bootstrapping test by a Welch's t-test performed for each bin. This method shows to be stricter than our initial bootstrapping method, and we are confident that it is more robust. We have altered the method section accordingly.

Line 255: How were these bins chosen? Are they representative of percentiles or thresholds that impacts users are concerned about? They seem random to me, and 60mm/day seems quite high for the high category for some regions.

We agree that the chosen values were not appropriate for each season and region. We have now changed the intervals to relative intervals defined for each region and season using the observational datasets (see Fig. 1). The low intensity interval now represents the lowest 40% of observational mean precipitation. The moderate intensity interval represents the middle bins cumulating the next 50% of mean precipitation. The high intensity interval represents the highest 10% of mean precipitation. We added vertical bars in the graphs to show the chosen intervals for each season and region.
This new representation changes our results (see new Fig. 8), mainly in the highest intervals, which now include more bins than with the fixed threshold (especially in winter). Our main conclusions remain similar. We re-wrote our results section with these new graphs which we believe are more robust.

Line 270: you should state here if they are in the supplement or not shown.

We have removed this sentence as we discuss all seasons with the pie-charts.

Fig 4, and other similar figures. The legend placement is sloppy in some of these images. It should be placed where it doesn't cover data.

Thank you, this has been fixed.

Line 330: I disagree with the statement that "there is no systematic difference". I see that there is a distinct systematic difference in most regions regardless of season, and so do you, clearly, because you discuss it in the next paragraph. P clearly has more light rain, C has more heavy rain, and the area under the curve is greater in C than P.

Please be more accurate.

We removed this sentence and rephrased the whole paragraph.

Line 336: the introduction to this sentence is confusing. Do you mean to state that "PRIMAVERA still overestimates low intensity precipitation in all seasons and regions, like CMIP5, although to a lesser extent."? And if so, where's the figure that shows that? I can't tell that it is "to a lesser extent" from the current content.

We have removed this sentence. We have also now replaced the CMIP5 figures by Fig. 2 and 3 that show CMIP5, EUR-11 and PRIMAVERA.

Line 340: At this point in the text, the statement here is not proven. You should add that this will be discussed later.

Agreed. We removed this sentence and have considerably changed the text.

Line 345: What is the sensitivity to the results in this paragraph to undercatch error?

See reply to comment L381 below.

Fig. 7. The C and P indicators on this figure are unfortunately too small for the resolution/quality of the PDF, and they are very hard to read, even when zoomed in, because they are so fuzzy. This needs to be fixed.

We fixed this by having only 4 panels instead of 9 in this figure. It is now Fig. S8.

Line 381: A pie chart including the observational uncertainty would be an interesting addition to the supplementary material, and address my question at line 345.

Thank you for this suggestion. We have included this new figure (Fig. S7). It shows that CORDEX intense precipitation is closer to the observations scaled by a factor 1.2. This is now discussed in a section on observational uncertainty (4.3).

Line 415: Given how similar PRIMAVERA and CORDEX-44 are, this statement is diffi- cult to verify by just eye-balling the difference between this figure and previous figures. CORDEX-44 and CORDEX-11 are not identical and some of the small differences may matter. Could you overlay all 3 datasets?

We have now redone our analyses with both CORDEX-44 and CORDEX-11, with a focus on CORDEX-11, which is a newer ensemble. We have created new figures showing seasonal mean bias (Fig. 4) and Taylor diagrams for all ensembles (Fig. 5). We also now include a pie chart for both CORDEX-44 and CORDEX-11 (Fig. 8), so the differences between CORDEX-11, CORDEX-44 and PRIMAVERA are clearly identified.

Line 445: Not all convective precipitation schemes (CPS) take into account convective inhibition in their triggering function. This makes a difference in the drizzle problem. In my experience, RCMs are more likely to use CPS that include this (in mid-latitudes at least), and GCMs are less likely to. This isn't tuning, it's a configuration choice.

We reworded the "tuning" expression, but we did not speculate further on this issue.

Line 483: "over CMIP5-driven CORDEX simulations for precipitation over Europe". . . this is a bit general. It would be more accurate to say "over CMIP5-driven CORDEX simulations in some regions and seasons by our metrics for precipitation over Europe"

Done.

Line 486: Please re-write this sentence, I do not understand at all what it is trying to say. And, if I guess at what it is trying to say, I do not agree with the statement.

The sentence has been reworded as 'It indicates that the potential improvement of large-scale dynamics in high-resolution GCMs has a positive influence on precipitation distribution.'

Technical Corrections:

I do not think it should be the responsibility of the reviewer to copy-edit the manuscript. This paper is readable and well-organized, but could use some English language copy- editing. I have pointed out important instances of text in the previous section that I think need to be corrected for clarity purposes though, as these effect the understanding of the science presented. Please pay particular attention those.

We have now carefully revised the manuscript.